# Online Change Point Detection for
# Multivariate Inhomogeneous Poisson Processes Time Series

Xiaokai Luo [1]   Haotian Xu [2]   Carlos Misael Madrid Padilla [3]   Oscar Hernan Madrid Padilla [4]

## Abstract

We study online change point detection for multivariate inhomogeneous Poisson point process time series. This setting arises commonly in applications such as earthquake seismology, climate monitoring, and epidemic surveillance, yet remains underexplored in the machine learning and statistics literature. We propose a method that uses low-rank matrices to represent the multivariate Poisson intensity functions, resulting in an adaptive nonparametric detection procedure. Our algorithm is single-pass and requires only constant computational cost per new observation, independent of the elapsed length of the time series. We provide theoretical guarantees to control the overall false alarm probability and characterize the detection delay under temporal dependence. We also develop a new Matrix Bernstein inequality for temporally dependent Poisson point process time series, which may be of independent interest. Numerical experiments demonstrate that our method is both statistically robust and computationally efficient.

## 1. Introduction

An inhomogeneous Poisson point process (PPP) provides a flexible model for random events occurring in space with location-dependent rates. Applications include forest fires (Stoyan & Penttinen, 2000), earthquakes (Bray & Schoenberg, 2013), citywide crime incidents (Baddeley et al., 2021), and epidemic outbreaks (Al-Dousari et al., 2021). In these examples, the *intensity function* is cen-

[1]Department of Applied and Computational Mathematics and Statistics, University of Notre Dame, Indiana, USA [2]Department of Mathematics and Statistics, Auburn University, Alabama, USA [3]Department of Statistics and Data Science, Washington University in St. Louis, Missouri, USA [4]Department of Statistics and Data Science, University of California, Los Angeles, USA. Correspondence to: Xiaokai Luo <xluo4@nd.edu>.

*Proceedings of the 43rd International Conference on Machine Learning*, Seoul, South Korea. PMLR 306, 2026. Copyright 2026 by the author(s).

tral, as it specifies the expected event density and encodes the spatial structure of the phenomenon. Formally, a PPP sample $X \subset \mathbb{R}^d$ is said to be sampled from an intensity $\lambda : \mathbb{R}^d \to \mathbb{R}_+$ if:

**(1)** for every set $S \subseteq \mathbb{R}^d$, the count $|S \cap X|$ is a Poisson random variable with mean $\int_S \lambda(x)\,dx$; and

**(2)** for disjoint sets $S_1, \ldots, S_n \subset \mathbb{R}^d$, the random variables $|S_1 \cap X|, \ldots, |S_n \cap X|$ are independent.

Estimation for a *single* PPP is well studied: Reynaud-Bouret (2003) derive minimax rates of estimating the intensity function in one dimension; Flaxman et al. (2017) estimate intensities using reproducing kernel Hilbert space (RKHS); and Xu et al. (2025) leverage tensor structure for multivariate intensity estimation.

In this work, we study online change point detection for PPP time series. Specifically, we assume that at every instance $i$, we observe a PPP sample $X^{(i)}$, and the underlying marginal intensity function can changes over time in an abrupt manner. A key feature of real data is that consecutive samples $X^{(i)}$ are temporally dependent. For example, environmental persistence and latent dynamics can induce temporal dependence across time points even when each sample is well approximated by a PPP marginally (Baddeley, 2007). To model this feature, we allow the sequence $\{X^{(i)}\}_{i=1}^{\infty}$ to exhibit temporal dependence, while maintaining PPP structure within each time index $i$. We formalize this general temporal dependent time series models below.

**Definition 1.1** (PPP time series with temporal dependence)**.** Let $\mathbb{X} \subset \mathbb{R}^d$ be compact. Let $\{X^{(i)}\}_{i=1}^{\infty}$ be a sequence of point processes on $\mathbb{X}$. Assume the following two conditions hold.

**(A) Poisson marginals.** For each time index $i$, the marginal distribution of $X^{(i)}$ corresponds to an inhomogeneous Poisson point process on $\mathbb{X}$ with intensity $\lambda_i^* : \mathbb{X} \to \mathbb{R}_+$, satisfying

$$\sup_{i \in \mathbb{Z}} \|\lambda_i^*\|_\infty < \infty \quad \text{and} \quad \sup_{i \in \mathbb{Z}} \|\lambda_i^*\|_{W^{2,\gamma}} < \infty,$$

where $\|\cdot\|_{W^{2,\gamma}}$ is the Sobolev norm defined in (1).

**(B) Geometric $\beta$-mixing across time.** There exists a con-

stant $c > 0$ such that

$$\beta(k) \leq e^{-c(k-1)} \qquad \text{for all } k = 1, 2, 3, \ldots .$$

Here, $\{\beta(k)\}_{k=1}^{\infty}$ are the $\beta$-mixing coefficients of $\{X^{(i)}\}_{i=1}^{\infty}$, as defined in (2).

Under **(A)** and **(B)**, we consider the possible two scenarios.

**(M0) No change point.** There exists an intensity $\lambda^* : \mathbb{X} \to \mathbb{R}_+$ such that

$$\lambda_i^* = \lambda^* \text{ for all } i \in \mathbb{Z}.$$

**(M1) Single change point.** There exist intensities $\lambda^*, \lambda_a^* : \mathbb{X} \to \mathbb{R}_+$ with $\lambda^* \neq \lambda_a^*$ and a change point $\mathfrak{b} \in \mathbb{Z}_+$ such that

$$\lambda_i^* = \lambda^* \text{ for all } i \leq \mathfrak{b}, \quad \text{and} \quad \lambda_i^* = \lambda_a^* \text{ for all } i > \mathfrak{b}.$$

While real-world time series may contain multiple change points, we follow the online change point detection literature and focus on the above at most one change point settings. In practice, the detection algorithm is simply restarted immediately after a change is declared.

## 1.1. Related Works

Nonparametric change detection for general distributions has been studied through a range of modern approaches. Representative lines of work include kernel methods that embed distributions into reproducing kernel Hilbert spaces and enable sequential two-sample testing (Harchaoui et al., 2008; Li et al., 2015; Arlot et al., 2019; Wei & Xie, 2026), discrepancy measures based on energy distances and characteristic functions (Matteson & James, 2014), and learning-based procedures such as neural network detectors (Li et al., 2024; Gong et al., 2022). Additional perspectives include functional kernel approaches (Romano et al., 2023), random forest methods (Londschien et al., 2023), and graph-based two-sample statistics (Chen & Zhang, 2015; Chu & Chen, 2019). Anytime-valid methodology via e-values has also been emphasized recently, providing online change detectors with rigorous error control under minimal assumptions (Shin et al., 2024). From a theoretical standpoint, minimax-optimal results for offline nonparametric change point detection and localization have been established for changes in smooth distributions (Madrid Padilla et al., 2021; 2023).

Change point detection for point process time series has wide-ranging applications, including earthquake seismology (Ogata, 2011), wildfire monitoring (Xu & Schoenberg, 2011), epidemic surveillance (Hohl et al., 2020) and DNA sequencing (Zhang et al., 2016). To the best of our knowledge, existing point process change point methods are largely tailored to parametric temporal or spatio-temporal models, or to offline estimation, and therefore are not directly applicable to our online nonparametric multivariate

PPP-intensity setting. In particular, Wang et al. (2023) consider sequential detection for self- and mutually-exciting point processes (specifically, Hawkes networks) using a parametric CUSUM/likelihood-based construction on temporal event data. Similarly, Zhang et al. (2023) study online score statistics for clustered changes in multivariate Hawkes network point processes. The composite-likelihood approach of Zhao et al. (2024) is also model-based and focuses on offline change-point estimation in piecewise stationary spatio-temporal processes. Finally, Dion-Blanc et al. (2023) study multiple *offline* change-point detection for some point processes, including inhomogeneous and marked Poisson processes, using a minimum-contrast estimator. Complementing algorithmic developments, Brandenberger et al. (2025) study fundamental detection limits for point-process changes from an information-theoretic perspective.

Despite these advances, existing change point methods for point processes often rely on strong parametric assumptions and typically assume independence across time. Our PPP time series setting poses additional challenges: each observation is an unordered random set with random cardinality, temporal dependence is present over time, and the signal corresponds to a change in an intensity function rather than a finite-dimensional parameter. To our knowledge, there is no general-purpose approach with provable guarantees for detecting nonparametric changes in multivariate PPP intensities under realistic temporal dependence.

## 1.2. Summary of Results

**New algorithm for online change detection.** We introduce a new computationally efficient online nonparametric detection procedure for PPP time series. The key idea is to map each PPP sample $X^{(i)}$ to a low-rank intensity matrix yielding a scalable algorithm adaptive to local changes in the underlying intensity. In particular, the per new observation cost is constant independent of the elapsed time series length. As a result, our method is single-pass, and the total computational cost scales linearly with the number of observed samples.

**General theory.** We develop a theoretical framework for online change detection in multidimensional PPPs under temporal dependence. A key technical ingredient is a new Matrix Bernstein inequality for geometrically $\beta$-mixing PPP time series (Theorem C.9 in Appendix). Combined with our low-rank intensity matrix approximation analysis, this yields sharp non-asymptotic bounds that simultaneously account for basis-truncation bias and stochastic variance in dependent PPP time series.

**Finite-sample guarantees.** We establish finite-sample guarantees to show that our newly proposed method both controls the overall false alarm probability and can detect a true change within a delay that depends explicitly on the change

size in the intensity functions.

**Empirical evidence.** We demonstrate that our procedure reliably identifies meaningful intensity changes while remaining computationally efficient through extensive simulations and a real-data application in modeling earthquake activity.

### 1.3. Notations

**Matrices.** For $\mathcal{M} \in \mathbb{R}^{m \times n}$, let $\mathcal{M}_{(\mu,\eta)}$ be its $(\mu, \eta)$ entry. We write $\|\mathcal{M}\|_{\mathrm{F}}$ and $\|\mathcal{M}\|_{\mathrm{op}}$ for the Frobenius and operator norms, and $\mathrm{Rank}(\mathcal{M})$ for the rank. If $\mathrm{Rank}(\mathcal{M}) = s$, let $\sigma_1(\mathcal{M}) \geq \cdots \geq \sigma_s(\mathcal{M}) > 0$ be its nonzero singular values. With the SVD $\mathcal{M} = U\Sigma V^\top$, define the rank-$r$ truncation $(r \leq s)$ by

$$\mathrm{SVD}(\mathcal{M}, r) = U \, \Sigma_{(r)} \, V^\top.$$

Here $\Sigma_{(r)} = \mathrm{diag}(\sigma_1(\mathcal{M}), \ldots, \sigma_r(\mathcal{M}), 0, \ldots, 0) \in \mathbb{R}^{s \times s}$.

**Function spaces.** Let $\mathbb{X} = \mathbb{X}_1 \times \cdots \times \mathbb{X}_d \subset \mathbb{R}^d$ be compact and define

$$\mathbf{L}_2(\mathbb{X}) = \big\{ f : \mathbb{X} \to \mathbb{R} : \|f\|_{\mathbf{L}_2}^2 = \int_{\mathbb{X}} f^2(x) \, \mathrm{d}x < \infty \big\}.$$

For convenience we often take $\mathbb{X}_1 = \cdots = \mathbb{X}_d = \Omega$. A family $\{\phi_k\}_{k \geq 1} \subset \mathbf{L}_2(\Omega)$ is orthonormal if

$$\int_\Omega \phi_k(x)\phi_j(x) \, \mathrm{d}x = \mathbf{1}\{k = j\}.$$

For an integer $\gamma \geq 1$, let $W^{2,\gamma}(\mathbb{X})$ be the Sobolev space of functions with weak derivatives $f^{(b)} \in \mathbf{L}_2(\mathbb{X})$ for all multi-indices $b$ with $|b| \leq \gamma$, equipped with the norm

$$\|f\|_{W^{2,\gamma}}^2 = \sum_{|b| \leq \gamma} \|f^{(b)}\|_{\mathbf{L}_2}^2. \tag{1}$$

**Temporal dependence ($\beta$-mixing).** Let $\{X_i\}_{i=-\infty}^{\infty}$ be a time series and define $\mathcal{F}_{-\infty}^j = \sigma(X_i : i \leq j)$ and $\mathcal{F}_{j+k}^\infty = \sigma(X_i : i \geq j + k)$ for $j \in \mathbb{Z}$ and $k \geq 1$. The $\beta$-mixing coefficients are

$$\beta(k) = \sup_{j \in \mathbb{Z}} \beta\big(\mathcal{F}_{-\infty}^j, \mathcal{F}_{j+k}^\infty\big), \qquad k \geq 1, \tag{2}$$

where, for two $\sigma$-fields $\mathcal{A}$ and $\mathcal{B}$, the absolute regularity coefficient is defined by

$$\beta(\mathcal{A}, \mathcal{B}) = \frac{1}{2} \sup \bigg\{ \sum_{i=1}^{I} \sum_{j=1}^{J} |\mathbb{P}(A_i \cap B_j) - \mathbb{P}(A_i)\mathbb{P}(B_j)| :$$
$$\{A_i\}_{i=1}^{I} \subset \mathcal{A}, \{B_j\}_{j=1}^{J} \subset \mathcal{B} \text{ are partitions} \bigg\}.$$

We refer interested readers to Bradley (2005) for a detailed discussion of mixing conditions.

## 2. Online Change Point Detection for Poisson Point Processes

We now describe the time series model and our detection procedure. We work under the PPP time series models with temporal dependence introduced in Definition 1.1.

**Training stage.** We observe a collection of point processes

$$X^{(i)} \subset \mathbb{X} = \Omega^{\otimes d} \subset \mathbb{R}^d, \quad i = 1, \ldots, N_{\mathrm{train}},$$

where each $X^{(i)}$ is generated over a fixed observation window (e.g., one day of events, one spatial snapshot, or one short space–time interval). We focus on the multivariate settings with $d \geq 2$. The case $d = 1$ is discussed in Remark 2.3. We assume $\{X^{(i)}\}_{i=1}^{N_{\mathrm{train}}}$ to follow (**M0**) in Definition 1.1 with pre-change intensity $\lambda^*$. The training sample is used to calibrate tuning parameters and the detection threshold.

**Post-training stage.** After the training stage, the sequence evolves according to one of two scenarios. In the first scenario (**M0**), the Poisson intensity remains unchanged for the rest of the time horizon; that is, $\{X^{(i)}\}_{i=1}^{\infty}$ is stationary with common intensity function $\lambda^*$. In the second scenario (**M1**), there exists an unknown change point $\mathfrak{b} \geq N_{\mathrm{train}}$ such that $\{X^{(i)}\}_{i=N_{\mathrm{train}}+1}^{\mathfrak{b}}$ is stationary with intensity $\lambda^*$ and $\{X^{(i)}\}_{i=\mathfrak{b}+1}^{\infty}$ is stationary with intensity $\lambda_a^*$. We allow temporal dependence within each regime under a geometric $\beta$-mixing condition, as specified in Definition 1.1 (**A**).

**Goal.** Given the training and post-training data, our goal is to develop an online algorithm such that (i) when there is no change, the overall probability of a false alarm is kept small; and (ii) if a change occurs after the training phase, the algorithm raises an alarm as quickly as possible to minimize the detection delay.

**Mapping intensity functions to matrices.** We represent the infinite-dimensional intensity functions to a finite-dimensional matrix via a RKHS basis. This representation has a distance-preserving property, enabling efficient operation over arbitrary intervals via dynamic programming, and avoids additional Monte Carlo procedures to compute $\mathbf{L}_2$ norms of functions in higher dimensions.

Formally, let $x = (x_1, \ldots, x_d) \in \mathbb{X} = \Omega^{\otimes d} \subset \mathbb{R}^d$, and partition the index set $[d] = \{1, \ldots, d\}$ into two disjoint subsets $\mathcal{I}_1$ and $\mathcal{I}_2$ such that

$$[d] = \mathcal{I}_1 \cup \mathcal{I}_2, \qquad |\mathcal{I}_1| = p, \ |\mathcal{I}_2| = q, \ p + q = d.$$

Define the corresponding coordinate partition as

$$y = (x_j)_{j \in \mathcal{I}_1} \in \Omega^{\otimes p}, \qquad z = (x_j)_{j \in \mathcal{I}_2} \in \Omega^{\otimes q},$$

we write $x = (y, z) \in \Omega^{\otimes p} \times \Omega^{\otimes q} \subset \mathbb{R}^{p+q}$.

Let $\{\phi_k\}_{k=1}^{\infty}$ form orthonormal univariate basis of $\mathbf{L}_2(\Omega)$. Then $\big\{\phi_{i_1}(x_1) \cdots \phi_{i_p}(x_p)\big\}_{i_1, \ldots, i_p=1}^{\infty}$ is a set of complete

basis functions of $\mathbf{L}_2(\Omega^{\otimes p})$. For any $M \in \mathbb{Z}_+$, the collection of functions $\{\phi_{i_1}(x_1) \cdots \phi_{i_p}(x_p)\}_{i_1,\dots,i_p=1}^M \subset \mathbf{L}_2(\Omega^{\otimes p})$ is orthonormal in $\Omega^{\otimes p}$ with cardinality $M^p$. Ordering the multi-indices $(i_1,\dots,i_p)$ and $(\ell_1,\dots,\ell_q)$, we denote

$$
\begin{aligned}
\left\{\Phi_\mu(y)\right\}_{\mu=1}^{M^p} &= \left\{\phi_{i_1}(x_1)\cdots\phi_{i_p}(x_p)\right\}_{i_1,\dots,i_p=1}^M, \\
\left\{\Psi_\eta(z)\right\}_{\eta=1}^{M^q} &= \left\{\phi_{\ell_1}(x_{p+1})\cdots\phi_{\ell_q}(x_{p+q})\right\}_{\ell_1,\dots,\ell_q=1}^M.
\end{aligned}
\tag{3}
$$

Let $\lambda^* : \Omega^{\otimes p} \times \Omega^{\otimes q} \to \mathbb{R}_+$ satisfy $\|\lambda^*\|_{W^{2,\gamma}} < \infty$. Define the matrix $\mathcal{M}(\lambda^*) \in \mathbb{R}^{M^p \times M^q}$ by

$$
\mathcal{M}(\lambda^*)_{(\mu,\eta)} = \iint_{\mathbb{R}^{p+q}} \lambda^*(y,z)\,\Phi_\mu(y)\,\Psi_\eta(z)\mathrm{d}y\mathrm{d}z. \tag{4}
$$

Using $\mathcal{M}(\lambda^*)$, we can approximate $\lambda^*$ by its truncated expansion:

$$
\lambda_M^*(y,z) = \sum_{\mu=1}^{M^p}\sum_{\eta=1}^{M^q} \mathcal{M}(\lambda^*)_{\mu,\eta}\,\Phi_\mu(y)\,\Psi_\eta(z). \tag{5}
$$

It was shown in Appendix G.1 that if $\{\phi_k\}_{k=1}^\infty$ are univariate RKHS orthonormal basis function, then

$$
\|\lambda^* - \lambda_M^*\|_{\mathbf{L}_2} \le C\,\|\lambda^*\|_{W^{2,\gamma}}\,M^{-\gamma}. \tag{6}
$$

*Remark* 2.1 (Coordinate split). It follows from (5) and (6) that $\mathcal{M}(\lambda^*)$ provides an accurate matrix representation of $\lambda^*$ with small approximation error. This representation requires a coordinate partition in $\mathbb{X} \subset \mathbb{R}^d$. As suggested by Theorem B.1 in Appendix, such a split is valid for any function that admits a functional PCA representation. The split can also be specified using prior knowledge of the dataset, as demonstrated in our real-data example. As a third option, one can partition the features into two groups so that variables are more correlated within groups and less correlated across groups. See Section 3.1 for more details.

## 2.1. Online change point detection

For each process $X^{(i)}$, define its intensity matrix by $\widehat{\mathcal{M}}^{(i)} \in \mathbb{R}^{M^p \times M^q}$ with entries

$$
\widehat{\mathcal{M}}^{(i)}_{(\mu,\eta)} = \sum_{x^{(i)}=(y^{(i)},z^{(i)})\in X^{(i)}} \Phi_\mu(y^{(i)})\,\Psi_\eta(z^{(i)}). \tag{7}
$$

It follows from Campbell's Theorem (Theorem C.4 in Appendix) that $\mathbb{E}(\widehat{\mathcal{M}}^{(i)}_{(\mu,\eta)}) = \mathcal{M}(\lambda^*)_{(\mu,\eta)}$. Therefore, under the single change point scenario **(M1)**, the intensity functions admit a change at $\mathfrak{b}$, and

$$
\mathbb{E}\big(\widehat{\mathcal{M}}^{(i)}\big) = \begin{cases} \mathcal{M}(\lambda^*) & \text{if } i \le \mathfrak{b}, \\ \mathcal{M}(\lambda_a^*) & \text{if } i > \mathfrak{b}. \end{cases} \tag{8}
$$

Consequently, for any $n \le \mathfrak{b}$, the deviation between the matrices $n^{-1}\sum_{i=1}^n \widehat{\mathcal{M}}^{(i)}$ and $\mathcal{M}(\lambda^*)$ can be controlled by high probability bounds in the time series setting.

Our online change detection procedure is summarized in Algorithm 1. Below we briefly explain its implementation. Using dynamic programming, at the current time $j$ and for any $k \in \{1,\dots,W\}$, with the window size $W$ such that $N_{\text{train}} + W \le j$, we maintain

$$
L[k] = \sum_{i=1}^{(j-W+k)-1} \widehat{\mathcal{M}}^{(i)} \quad \text{and} \quad R[k] = \sum_{i=(j-W+k)}^{j} \widehat{\mathcal{M}}^{(i)}.
$$

Hence, for a given pair $(j,k)$, the matrix $\mathcal{D} \in \mathbb{R}^{M^p \times M^q}$ in Algorithm 1 is the CUSUM statistic

$$
\begin{aligned}
\mathcal{D} = {}& \frac{1}{(j-W+k)-1} \sum_{i=1}^{j-W-1+k} \widehat{\mathcal{M}}^{(i)} \\
& - \frac{1}{W-k+1} \sum_{i=(j-W+k)}^{j} \widehat{\mathcal{M}}^{(i)},
\end{aligned}
\tag{9}
$$

which compares the data between the intervals $[1,(j-W+k)-1]$ and $[(j-W+k),j]$. For example if $j = \mathfrak{b} + W$ and $k = 1$, from (8) we can deduce that

$$
\begin{aligned}
\mathcal{D} &= \frac{1}{\mathfrak{b}}\sum_{i=1}^{\mathfrak{b}} \widehat{\mathcal{M}}^{(i)} - \frac{1}{W}\sum_{i=\mathfrak{b}+1}^{\mathfrak{b}+W} \widehat{\mathcal{M}}^{(i)} \\
&\approx \mathcal{M}(\lambda^*) - \mathcal{M}(\lambda_a^*) = \mathcal{M}(\lambda^* - \lambda_a^*),
\end{aligned}
\tag{10}
$$

where the last equality follows from the linearity of the coefficient/matricization operator $\mathcal{M}$ given in (4), that is, $\mathcal{M}(f-g) = \mathcal{M}(f) - \mathcal{M}(g)$ for any square-integrable functions $f$ and $g$. To further reduce variance when estimating $\lambda^* - \lambda_a^*$, we apply the restricted SVD procedure to $\mathcal{D}$ as described in Algorithm 2.

Algorithm 2 has two components: (i) zeroing out higher-order entries of $\mathcal{D}$ by trimming to an adaptive basis size, and (ii) applying SVD to the trimmed matrix. The trimming is adaptive to the sample size in (9): as $k$ ranges from 1 to $W$, the effective sample size is $W-k+1$. The necessity of trimming comes from the fact that smaller samples only allow us to reliably estimate a smaller number of matrix coefficients.

We apply SVD to $\mathcal{D}$ because its population counterpart $\lambda^* - \lambda_a^*$ is typically approximately low rank. Since $\lambda^* - \lambda_a^* \in \mathbf{L}_2(\Omega^{\otimes p} \times \Omega^{\otimes q})$, the functional SVD (Theorem B.1 in Appendix) yields

$$
\lambda^* - \lambda_a^* = \sum_{k=1}^{\infty} \sigma_k(\lambda^* - \lambda_a^*)\,f_k^*(y)\,g_k^*(z), \tag{11}
$$

with nonincreasing singular values $\sigma_1(\lambda^* - \lambda_a^*) \geq \sigma_2(\lambda^* - \lambda_a^*) \geq \cdots \geq 0$ such that

$$\sum_{k=1}^{\infty} \sigma_k^2(\lambda^* - \lambda_a^*) = \|\lambda^* - \lambda_a^*\|_{\mathbf{L}_2}^2 < \infty,$$

and orthonormal functions

$$\{f_k^*(y)\} \subset \mathbf{L}_2(\Omega^{\otimes p}), \quad \{g_k^*(z)\} \subset \mathbf{L}_2(\Omega^{\otimes q}).$$

It is a commonly used assumption in the literature (e.g., Hall et al., 2006; Raskutti et al., 2012) that, if $\|\lambda^* - \lambda_a^*\|_{W^{2,\gamma}} < \infty$, then the singular values of $\lambda^* - \lambda_a^*$ decay at a polynomial or exponential rate. Since $\mathcal{M}(\lambda^* - \lambda_a^*)$ provides an accurate matrix representation of $\lambda^* - \lambda_a^*$, we can anticipate that the singular values of $\mathcal{M}(\lambda^* - \lambda_a^*)$, and consequently the singular values of $\mathcal{D}$ in (10), decay at the same rate, see Lemma C.5 in Appendix for a justification.

*Remark* 2.2 (Computational cost). Due to the dynamic programming design, for a new observation the computational cost of Algorithm 1 is $O\big(rW^{1+d/(2\gamma+p\vee q)}\big)$. More precisely, at time $j$, updating each matrix in the lists $L$ and $R$ costs $O\big(W^{d/(2\gamma+p\vee q)}\big)$. Computing the rank-$r$ SVD for each difference matrix $\mathcal{D}$ in Algorithm 1 costs $O\big(rW^{d/(2\gamma+p\vee q)}\big)$. Since $L$ and $R$ each contain $W$ matrices, the total cost is $O\big(rW^{1+d/(2\gamma+p\vee q)}\big)$. Consequently, the method is single-pass over the sequential data, and the cost per new sample does not grow with the past time series length.

*Remark* 2.3 (PPP change point detection in 1D). Algorithm 1 tackles online change point detection for Poisson point process sequential data in $\mathbb{X} \subset \mathbb{R}^d$ with $d \geq 2$. On the other hand, in Section D of Appendix, we present a simplified one-dimensional variant that handles PPP time series in $\mathbb{R}$ by representing the intensity as a vector rather than a matrix. The univariate setting is substantially simpler than the multivariate setting, as univariate nonparametric models do not suffer from the curse of dimensionality.

In Theorem 2.4, we provide statistical guarantees for the overall false alarm probability and the detection delay of Algorithm 1.

**Theorem 2.4** (False-alarm control and detection delay). *Let the univariate basis functions $\{\phi_k\}_{k=1}^{\infty}$ in (3) be the Legendre polynomials. Assume the PPP time series $\{X^{(i)}\}_{i=1}^{\infty}$ satisfies Definition 1.1 on a compact domain $\mathbb{X} \subset \mathbb{R}^d$, with $d = p + q \geq 2$. Suppose the training length $N_{\text{train}}$ is sufficiently large.*

*(a) No change point. Under scenario (M0) with intensity $\lambda^*$. Choosing a sufficient large threshold constant $\mathcal{C}_\alpha$ in Algorithm 1, with probability at least $1 - \alpha$, Algorithm 1 never raises an alarm over the entire time horizon.*

**Algorithm 1** Online multivariate PPP change detection

**Input:** Smoothness parameter $\gamma > 0$; dimensionality $p, q$ with $p + q = d$; rank $r$; window size $W$; threshold constant $\mathcal{C}_\alpha$

▸ **Initialization Stage**
$M \leftarrow \lceil (W/r)^{1/(2\gamma + p\vee q)} \rceil$
**for** $k = 1$ **to** $W$ **do**
   $L[k] \leftarrow \sum_{i=1}^{N_{\text{train}} - W + k - 1} \widehat{\mathcal{M}}^{(i)} \in \mathbb{R}^{M^p \times M^q}$   (computed via (7))
**end for**

▸ **Detection Stage**
ALARM $\leftarrow$ FALSE
**for** $j = N_{\text{train}} + 1, N_{\text{train}} + 2, \ldots$ **do**
  **for** $k = 1$ **to** $W - 1$ **do**
    $L[k] \leftarrow L[k+1]$
  **end for**
  $L[W] \leftarrow L[W] + \widehat{\mathcal{M}}^{(j-1)}$
  **for** $k = 1$ **to** $W$ **do**
    $R[k] \leftarrow \sum_{i=j-W+k}^{j} \widehat{\mathcal{M}}^{(i)}$
  **end for**
  **for** $k = 1$ **to** $W$ **do**
    $n_1 \leftarrow j - W - 1 + k$
    $n_2 \leftarrow W - k + 1$
    $\mathcal{D} \leftarrow n_1^{-1} L[k] - n_2^{-1} R[k]$   (defined in (9))
    **if** $\text{RSVD}(\mathcal{D}) > \mathcal{C}_\alpha \left(\frac{r}{n_2}\right)^{\frac{\gamma}{2\gamma + p\vee q}} \log(j)$   (see Algorithm 2) **then**
      ALARM $\leftarrow$ TRUE
      **break**
    **end if**
  **end for**
**end for**

*(b) Single change point. Under scenario (M1) with change point $\mathfrak{b} \geq N_{\text{train}}$ and intensities $\lambda^*, \lambda_a^*$. Let $\{\sigma_k(\lambda^* - \lambda_a^*)\}_{k \geq 1}$ be the singular values of $\lambda^* - \lambda_a^*$, as specified in (11), and choose the rank parameter $r$ in Algorithm 1 such that*

$$\sqrt{\sum_{k=r+1}^{\infty} \sigma_k^2(\lambda^* - \lambda_a^*)} \leq \frac{\|\lambda^* - \lambda_a^*\|_{\mathbf{L}_2}}{5}. \qquad (12)$$

*Let $\kappa = \|\lambda^* - \lambda_a^*\|_{\mathbf{L}_2}$ and define*

$$\Delta = \left\lceil C_{\text{lag}} r \left( \log(\mathfrak{b})/\kappa \right)^{2 + (p\vee q)/\gamma} \right\rceil, \qquad (13)$$

*where $C_{\text{lag}}$ is a sufficiently large constant depending only on $\mathcal{C}_\alpha$. If the window size satisfies $W \geq \Delta$, then with probability at least $1 - \mathfrak{b}^{-3}$, Algorithm 1 raises an alarm within the time interval $(\mathfrak{b}, \mathfrak{b} + \Delta)$.*

**Algorithm 2** Restricted SVD: RSVD($\mathcal{D}$)

**Input:** Matrix $\mathcal{D} \in \mathbb{R}^{M^p \times M^q}$; rank $r$; sample size $n_2$; dimensions $p, q$; smoothness $\gamma > 0$

▶ **Adaptive trimming**

$m \leftarrow \left\lceil (n_2/r)^{1/(2\gamma + p \vee q)} \right\rceil$

$\mathcal{B}_y \leftarrow \left\{ \phi_{i_1}(x_1) \cdots \phi_{i_p}(x_p) \right\}_{i_1,\ldots,i_p=1}^{m}$

$\mathcal{B}_z \leftarrow \left\{ \phi_{\ell_1}(x_{p+1}) \cdots \phi_{\ell_q}(x_{p+q}) \right\}_{\ell_1,\ldots,\ell_q=1}^{m}$

**for** $\mu = 1$ **to** $M^p$ **do**
  **if** $\Phi_\mu \notin \mathcal{B}_y$ **then**
    set the $\mu$-th row $\mathcal{D}_{\mu,*} \leftarrow 0$
  **end if**
**end for**
**for** $\eta = 1$ **to** $M^q$ **do**
  **if** $\Psi_\eta \notin \mathcal{B}_z$ **then**
    set the $\eta$-th column $\mathcal{D}_{*,\eta} \leftarrow 0$
  **end if**
**end for**

▶ **Rank-$r$ projection and score**
$\mathcal{D}[r] \leftarrow \text{SVD}(\mathcal{D}, r)$
**Output:** $\|\mathcal{D}[r]\|_{\text{F}}$

---

In Theorem 2.4 **(a)**, $\mathcal{C}_\alpha$ is the threshold parameter that controls the overall false-alarm probability. We discuss a data-driven strategy for calibrating $\mathcal{C}_\alpha$ in Section 3.1.

*Remark* 2.5 (Generality of Theorem 2.4 across coordinate partitions). The condition (12) in Theorem 2.4 follows from a commonly used functional regularity condition in the literature. In particular, we show in Lemma C.6 in the Appendix that, if the singular values $\{\sigma_k(\lambda^* - \lambda_a^*)\}_{k=1}^\infty$ decay at a polynomial (with degree $> 1/2$) or exponential rate, then (12) holds with a constant rank $r$. Such functional singular-value decay assumptions are standard in the literature (e.g., Hall et al., 2006; Raskutti et al., 2012), and they accommodate separable, additive, and finite-rank intensity classes as well as more general smooth interactions. The factor 5 in the denominator of (12) is chosen for convenience and we do not attempt to optimize it.

Importantly, Theorem 2.4 establishes the false-alarm and detection-delay guarantees for *any* coordinate partition $[d] = \mathcal{I}_1 \cup \mathcal{I}_2$ for which the matricized intensity satisfies (12); the proof does not rely on any specific property of the partition-selection rule. The empirical cross-group-correlation criterion in Section 3.1 is therefore a practical and numerically stable heuristic for locating such a partition rather than a structural requirement of the theorem; robustness to alternative rank choices and to random coordinate partitions is verified empirically in our numerical studies.

Theorem 2.4 also implies that, when a change point is

present, the detection delay is at most $O(\kappa^{-2-(p \vee q)/\gamma})$. By comparison, Madrid Padilla et al. (2021) and Madrid Padilla et al. (2023) show that, for nonparametric density change point detection in the offline setting, the error scales as $O(\kappa^{-2-d/\gamma})$, where $\kappa$ is the size of the change in $\mathbf{L}_2$-norm, $\gamma$ is the degree of smoothness, and $d$ is the ambient dimension of the density function. Although we study Poisson intensity changes for time series data, our detection delay bound is strictly better in order. This is because $p + q = d$, and thus $p \vee q < d$. In the difficult regime where the change size is small, $\kappa \to 0$, we have

$$\kappa^{-2-(p \vee q)/\gamma} \ll \kappa^{-2-d/\gamma}.$$

## 3. Numerical Studies

We designed simulated experiments to evaluate the performance of Algorithm 1, which we refer to as the **Matrix detector**, against two benchmarks. First, the **MMD detector** follows Li et al. (2015): it represents the data in each window via its empirical kernel mean embedding and computes a blockwise maximum mean discrepancy (MMD) statistic between the point samples from the pre- and post-change segments. Second, the **KIE detector** is adapted from the density change point detector of Madrid Padilla et al. (2023): it estimates the pre- and post-change intensities using a kernel intensity estimator (KIE) and then forms a CUSUM statistic based on the discrepancy between the two estimated intensities, measured in the $\mathbf{L}_2$-norm. A Python implementation of our method is available at the GitHub repository.

### 3.1. Selection of Tuning Parameters

**MMD detector.** For the MMD detector, we adopt the default tuning-parameter selection in Li et al. (2015) and use a Gaussian kernel throughout the numerical experiments.

**KIE detector.** We follow the default choice in Madrid Padilla et al. (2023), and select the kernel bandwidth as well as the detection threshold using cross-validation on the training data.

**Matrix detector.** We set the smoothness parameter $\gamma = 2$, meaning the intensity functions are at least twice differentiable, which is standard in the nonparametric literature (e.g., Wasserman, 2006). We partition the $d$ coordinates into two groups using a correlation-based split criterion. Let $\text{Corr}(i, j)$ denote the empirical correlation between the $i$-th and $j$-th coordinate of all points in the training data. For any partition $(A, B)$ of $\{1, \ldots, d\}$, define

$$\Delta(A, B) = \frac{1}{|A||B|} \sum_{i \in A} \sum_{j \in B} |\text{Corr}(i, j)|.$$

We select the coordinate partition used in Algorithm 1 by

searching over all nontrivial partitions $(A, B)$ and choosing

$$(A^*, B^*) \in \underset{(A,B)}{\arg\min} \ \Delta(A, B),$$

and then set $p = |A^*|$ and $q = |B^*|$ such that $p + q = d$.

Given the coordinate partition and hence $p$ and $q$, we next select the rank parameter $r$ using sample splitting and a goodness-of-fit criterion. Without loss of generality, assume $N_{\text{train}}$ is even. For any candidate rank $r$, define

$$\widehat{\mathcal{W}}_1 = \frac{2}{N_{\text{train}}} \sum_{i=1}^{N_{\text{train}}/2} \widehat{\mathcal{M}}^{(i)} \quad \text{and}$$

$$\widehat{\mathcal{W}}_2 = \frac{2}{N_{\text{train}}} \sum_{i=N_{\text{train}}/2+1}^{N_{\text{train}}} \widehat{\mathcal{M}}^{(i)},$$

where $\widehat{\mathcal{M}}^{(i)}$ is defined in (7). Let $\text{SVD}(\mathcal{W}, r)$ denote the rank-$r$ truncated SVD approximation of $\mathcal{W}$. We choose $r$ by

$$\widehat{r} \in \underset{r \in \{1,\ldots,\min(M^p, M^q)\}}{\arg\min} \left\| \text{SVD}(\widehat{\mathcal{W}}_1, r) - \text{SVD}(\widehat{\mathcal{W}}_2, r) \right\|_{\text{F}}.$$

The last parameter to select is the threshold $\mathcal{C}_\alpha$. We calibrate $\mathcal{C}_\alpha$ using a block-permutation procedure on the training data. Fix a block length $L_{\text{block}}$ and let $K_{\text{block}} = \lfloor N_{\text{train}}/L_{\text{block}} \rfloor$. We partition the first $K_{\text{block}} L_{\text{block}}$ training windows into consecutive blocks

$$\mathcal{B}_\ell = \{(\ell-1)L_{\text{block}}+1, \ldots, \ell L_{\text{block}}\}, \quad \ell = 1, \ldots, K_{\text{block}}.$$

For each repetition $b = 1, \ldots, 500$, we randomly permute the $K_{\text{block}}$ blocks and split the permuted blocks into two groups of sizes $\lfloor K_{\text{block}}/2 \rfloor$ and $K_{\text{block}} - \lfloor K_{\text{block}}/2 \rfloor$ (equal when $K_{\text{block}}$ is even, and differing by one block otherwise). Let $\mathcal{J}_{1,b}$ and $\mathcal{J}_{2,b}$ denote the indices of the first and second block groups, and define the corresponding time-index sets

$$\mathcal{I}_{1,b} = \bigcup_{\ell \in \mathcal{J}_{1,b}} \mathcal{B}_\ell, \qquad \mathcal{I}_{2,b} = \bigcup_{\ell \in \mathcal{J}_{2,b}} \mathcal{B}_\ell.$$

We then compute the corresponding two-sample CUSUM matrix

$$\widehat{\mathcal{D}}_b = \frac{1}{|\mathcal{I}_{1,b}|} \sum_{i \in \mathcal{I}_{1,b}} \widehat{\mathcal{M}}^{(i)} - \frac{1}{|\mathcal{I}_{2,b}|} \sum_{i \in \mathcal{I}_{2,b}} \widehat{\mathcal{M}}^{(i)},$$

and record the test statistic $\|\widehat{\mathcal{D}}_b\|_{\text{F}}$. Finally, we set $\mathcal{C}_\alpha$ to be the $(1 - \alpha)$-quantile of

$$\left\{ \frac{\|\widehat{\mathcal{D}}_b\|_{\text{F}}}{(\widehat{r}/(|\mathcal{I}_{1,b}| \wedge |\mathcal{I}_{2,b}|))^{\gamma/(2\gamma+p\vee q)} \log(N_{\text{train}})} \right\}_{b=1}^{500},$$

and use it as the input threshold in Algorithm 1.

*Table 1.* Simulation results in the 3D setting over 100 replications. False Alarm denotes an alarm raised at or before $\flat$, Correct Detection denotes an alarm raised in $(\flat, N_{\text{total}}]$, and No Alarm denotes no alarm by $N_{\text{total}}$. The average detection delay (ADD) and the corresponding standard deviation (SD) are reported conditional on correct detection.

| | Matrix | MMD | KIE |
|---|---|---|---|
| False Alarm | 6% | 5% | 6% |
| Correct Detection | 94% | 93% | 89% |
| No Alarm | 0% | 2% | 5% |
| ADD (SD) | 9.19 (3.80) | 40.39 (29.79) | 43.98 (34.63) |

### 3.2. 3D Intensity with Temporally Dependent Latent Variables

We generate Poisson point process time series data in $d = 3$ dimensions with a change point at time $\flat = 1200$. We sample each process window using the thinning algorithm (Lewis & Shedler, 1979). Specifically, for $t \leq \flat$, the intensity function is

$$\lambda_t^*(x) = z_{t,1}^+ \prod_{j=1}^{3} \{\sin(x_j) + 1\} + z_{t,2}^+ \prod_{j=1}^{3} \{\cos(x_j) + 1\},$$

while for $t > \flat$ the intensity function is

$$\lambda_t^*(x) = z_{t,1}^+ \prod_{j=1}^{3} \exp(-x_j^2) + z_{t,2}^+ \prod_{j=1}^{3} x_j.$$

Here $x \in [0,1]^3$, $z_t^+ = \max\{z_t, 0\}$, and $z_t = (z_{t,1}, z_{t,2}) \in \mathbb{R}^2$ follows an autoregressive model

$$z_{t+1} = \begin{bmatrix} 0.5 & 0.1 \\ 0.1 & 0.5 \end{bmatrix} z_t + \epsilon_t, \qquad \epsilon_t \overset{i.i.d.}{\sim} N([3, 1], I_2),$$

with $I_2$ the two-dimensional identity matrix. Each run consists of $N_{\text{train}} = 1000$ pre-change samples and $N_{\text{total}} = 1500$ samples in total. All approaches are initialized on the same training data and evaluated over 100 Monte Carlo replications. In Table 1, we summarize the performance of the three methods using the default tuning-parameter choices described in Section 3.1.

To further understand the performance of our proposed method, we evaluate 15 threshold values per method, ranging from low detection sensitivity to high. Performance was summarized using two metrics: (i) false alarm probability (FAP) and (ii) average detection delay (ADD). If no alarm occurs by $N_{\text{total}}$, we set the delay to $N_{\text{total}} - \flat$.

Figure 1 presents the trade-off between empirical FAP and ADD. Our method (**Matrix**) generally achieves a smaller average detection delay than the other two methods at comparable false alarm probabilities.

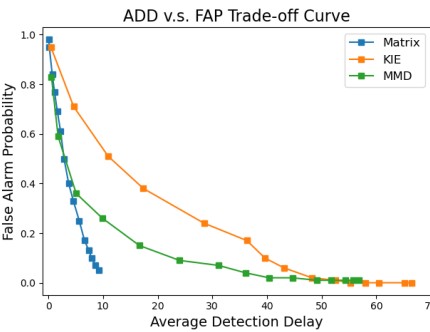

*Figure 1.* FAP vs. ADD comparison among three detectors under the 3D simulation setting.

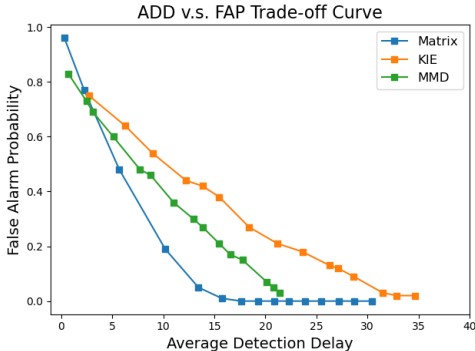

*Figure 2.* FAP vs. ADD comparison among three detectors under the 4 D simulation setting.

### 3.3. 4D Intensity with Autoregressive Scale

We generate Poisson point process time series data in $d = 4$ dimensions with a change point at time $\mathfrak{b} = 1200$. The intensity function is

$$\lambda_t^*(x) = y_t^+ \left\{ \prod_{j=1}^{4} 2x_j^3 + \prod_{j=1}^{4} 2\exp(-x_j) \right\}, \qquad x \in [0,1]^4,$$

where the scale sequence $\{y_t\}_{t=1}^{\infty}$ follow the autregressive model

$$y_{t+1} = \begin{cases} 0.1 \cdot |X^{(t)}| + 8 + \epsilon_t & \text{for } t \leq \mathfrak{b} \\ 0.1 \cdot |X^{(t)}| + 4 + \epsilon_t & \text{for } t > \mathfrak{b} \end{cases},$$

with $\epsilon_t \overset{i.i.d.}{\sim} N(0,1)$, and $y_t^+ = \max\{y_t, 0\}$. Here $X^{(t)}$ denotes the Poisson point process observed at time $t$ with intensity $\lambda_t^*$, and $|X^{(t)}|$ is its cardinality. Each run consists of $N_{\text{train}} = 1000$ pre-change samples and $N_{\text{total}} = 1500$ samples in total. All approaches were initialized on the same training data and evaluated over 100 Monte Carlo replications. In Table 2, we summarize the numerical performance of the three methods based on the default tuning-parameter choices described in Section 3.1.

*Table 2.* Simulation results in the 4D setting over 100 Monte Carlo replications. The average detection delay (ADD) and the corresponding standard deviation (SD) are reported conditional on correct detection.

|  | Matrix | MMD | KIE |
|---|---|---|---|
| False Alarm | 3% | 4% | 3% |
| Correct Detection | 97% | 94% | 92% |
| No Alarm | 0% | 2% | 5% |
| ADD (SD) | 13.44 (5.21) | 20.83 (10.90) | 31.50 (21.88) |

Figure 2 presents the trade-off between empirical false alarm probability (FAP) and average detection delay (ADD). Our method achieves substantially lower detection delays than the other two methods at comparable false alarm probabilities.

### 3.4. A Real Data Example

Poisson point processes are widely used to model earthquake activity (Anagnos & Kiremidjian, 1988; Cornell et al., 1968; Wang, 2006). We test our online change detection method on an earthquake dataset from the state of Oklahoma covering January 2000 to December 2018, obtained from the (U.S. Geological Survey, 2025). The dataset records each earthquake's longitude, latitude, depth, and magnitude, and we aggregate events by week so that each month forms one Poisson point process observation. We use data from January 2000 to December 2007 for training.

In this real-data experiment, we select all tuning parameters according to the procedure described in Section 3.1. Our online procedure raises an alarm in June 2009. This timing is consistent with reports by U.S. national agencies indicating that earthquake activity in Oklahoma, particularly for events with magnitude below 3, increased rapidly beginning around 2009 and has been linked to wastewater disposal and injection practices; see (U.S. Geological Survey, 2019). Figure 3 visualizes this trend through a time series of estimated yearly average intensity surfaces over six consecutive 12-month windows; the detected change point lies between the first two windows, and the seismic-activity hotspots expand markedly in the windows that follow. In contrast, the **KIE** detector raises an alarm in July 2007, while the **MMD** detector does not raise an alarm by December 2018.

### 4. Conclusion

We studied online change point detection for multivariate inhomogeneous Poisson point processes (PPP) time series under the $\beta$-mixing temporal dependence assumption. Our approach maps each PPP realization to a finite-dimensional intensity matrix via an orthonormal basis expansion, and then leverages approximate low-rank structure through a restricted SVD. This representation yields a single-pass detection algorithm with constant per-observation cost and

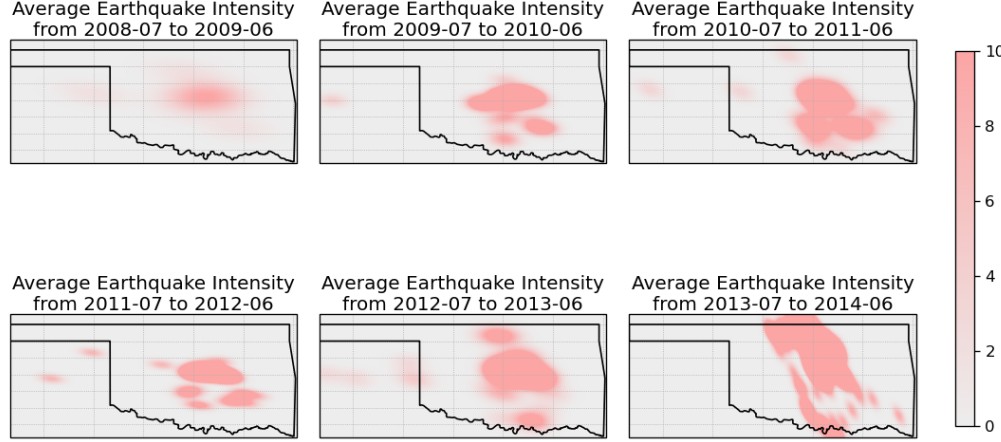

*Figure 3.* Estimated yearly average earthquake intensity in Oklahoma over six consecutive 12-month windows from July 2008 through June 2014. The change point detected by Algorithm 1 is June 2009, which lies between the first two panels. The panels reveal a clear and progressive increase in seismic activity in the years following the detected change.

total runtime linear in the number of windows. Theoretically, we develop a framework jointly capture the approximation bias and stochastic variance and provides explicit guarantees for both false-alarm control and detection delay. Empirically, our method demonstrates strong detection performance and robustness across a range of regimes, while maintaining computational efficiency that scales linearly in the time series length.

## Acknowledgements

We sincerely thank the reviewers and the Area Chair for their careful reading of our manuscript and for their constructive comments and suggestions, which has helped us substantially improve the manuscript.

## Impact Statement

This paper presents work whose goal is to advance the field of Machine Learning, specifically in the domain of online change point detection for multivariate point processes time series. Our proposed method has potential applications in areas such as seismology, where it can assist in monitoring earthquake intensity changes (as demonstrated in our experiments), as well as in network security and public health monitoring.

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

# A. Proof of Theorem 2.4

**Setup and notation.** Let $\{X^{(i)}\}_{i \geq 1}$ be a sequence of PPPs satisfying Definition 1.1 on a compact domain $\mathbb{X} \subset \mathbb{R}^d$. Let $\{\phi_k\}_{k \geq 1}$ be an orthonormal basis of $\mathbf{L}_2(\Omega)$ such that $\|\phi_k\|_\infty \leq C_\phi$ for all $k$, where $0 < C_\phi < \infty$ is an absolute constant. Fix integers $p, q \geq 1$ with

$$p + q = d \qquad \text{and} \qquad p \geq q.$$

For $x = (x_1, \ldots, x_d) \in \mathbb{X} = \Omega^{\otimes d}$, write the coordinate split

$$x = (y, z), \qquad y = (x_1, \ldots, x_p) \in \Omega^{\otimes p}, \qquad z = (x_{p+1}, \ldots, x_{p+q}) \in \Omega^{\otimes q}.$$

For an integer $m \geq 1$, define the tensor-product bases

$$\{\Phi_\mu(y)\}_{\mu=1}^{m^p} = \left\{ \phi_{\mu_1}(x_1) \cdots \phi_{\mu_p}(x_p) \right\}_{\mu_1,\ldots,\mu_p=1}^m,$$

$$\{\Psi_\eta(z)\}_{\eta=1}^{m^q} = \left\{ \phi_{\ell_1}(x_{p+1}) \cdots \phi_{\ell_q}(x_{p+q}) \right\}_{\ell_1,\ldots,\ell_q=1}^m.$$

For each window $i$ and basis size $m$, define the empirical intensity matrix $\widehat{A}^{(i),m} \in \mathbb{R}^{m^p \times m^q}$ entrywise by

$$\widehat{A}^{(i),m}_{(\mu,\eta)} = \sum_{x=(y,z) \in X^{(i)}} \Phi_\mu(y) \, \Psi_\eta(z).$$

Given integers $1 \leq t < n$, define the (two-sample) CUSUM matrix $\widehat{A}^m_{n,t} \in \mathbb{R}^{m^p \times m^q}$ by

$$\widehat{A}^m_{n,t} = \frac{1}{t} \sum_{i=1}^t \widehat{A}^{(i),m} - \frac{1}{n-t} \sum_{i=t+1}^n \widehat{A}^{(i),m}, \qquad \widehat{A}^m_{n,t}[r] = \mathrm{SVD}(\widehat{A}^m_{n,t}, r).$$

In our procedure, the basis size is chosen adaptively as a function of $(n, t)$:

$$m_{n,t} = \left\lceil \left( \frac{n-t}{r} \right)^{1/(2\gamma+p)} \right\rceil. \tag{14}$$

For brevity, we write

$$\widehat{A}_{n,t} = \widehat{A}^{m_{n,t}}_{n,t}, \qquad \widehat{A}_{n,t}[r] = \mathrm{SVD}(\widehat{A}_{n,t}, r). \tag{15}$$

Finally, define the threshold

$$\tau_{n,t} = C_\alpha \left( \frac{r}{n-t} \right)^{\gamma/(2\gamma+p)} \log n, \tag{16}$$

where $C_\alpha > 0$ is a sufficiently large constant depending only on $\alpha$.

*Proof of Theorem 2.4.* Theorem 2.4 directly follows Lemma A.1 and Lemma A.2. $\qquad \square$

**Lemma A.1.** *Let $m_{n,t}$, $\widehat{A}_{n,t}[r]$ and $\tau_{n,t}$ be defined in (14), (15) and (16), respectively. Suppose $\{X^{(i)}\}_{i=1}^\infty$ are sampled from (M0) with the common intensity function $\lambda^*$. Then*

$$\mathbb{P}\left( \|\widehat{A}_{n,t}[r]\|_{\mathrm{F}} \geq \tau_{n,t} \text{ for all } 1 \leq t < n < \infty \right) \leq 1 - \alpha.$$

*Proof.* Since there is no change points, we have $\delta^* = \lambda^* - \lambda_a^* = 0$. We apply Lemma B.4 to any give $t, n$ and have that with probability at most $2\epsilon n^{-3}$,

$$\|\widehat{A}_{n,t}[r]\|_{\mathrm{F}} \geq C_2 m_{n,t}^{-\gamma} + C_\epsilon \sqrt{r} \{\log^2(m_{n,t}) + \log^2(n)\} \sqrt{\frac{m_{n,t}^p}{n-t}},$$

where we have used $\sigma_r(\delta^*) = 0$ when $\delta^* = 0$. Since $m_{n,t} = \lceil (\frac{n-t}{r})^{1/(2\gamma+p)} \rceil$, it follows that

$$\mathbb{P}\left( \|\widehat{A}_{n,t}[r]\|_{\mathrm{F}} \geq C'_\epsilon \left( \frac{r}{n-t} \right)^{\gamma/(2\gamma+p)} \log^2(n) \right) \leq 2\epsilon n^{-3}.$$

By a union bound, for any $n \in \mathbb{Z}^+$ it holds that

$$\mathbb{P}\left( \|\widehat{A}_{n,t}[r]\|_{\mathrm{F}} \geq C'_\epsilon \left(\frac{r}{n-t}\right)^{\gamma/(2\gamma+p)} \log(n) \text{ for all } 1 \leq t < n \right) \leq 2\epsilon n^{-2}.$$

Since $\sum_{n=1}^\infty n^{-2} = \pi^2/6$, if follows that if $\epsilon = 3\alpha/\pi^2$, then by another union bound,

$$\mathbb{P}\left( \|\widehat{A}_{n,t}[r]\|_{\mathrm{F}} \geq C_\alpha \left(\frac{r}{n-t}\right)^{\gamma/(2\gamma+p)} \log(n) \text{ for all } 1 \leq t < n < \infty \right) \leq \alpha.$$

The desired result follows from the choice of $\tau_{n,t}$ in (16). $\qquad\square$

**Lemma A.2.** *Let $\tau_{n,t}$ be defined in (16). Suppose $\{X^{(i)}\}_{i=1}^{\mathfrak{b}} \cup \{X^{(i)}\}_{i=\mathfrak{b}+1}^{\infty}$ are sampled from (M1) with true change point $\mathfrak{b}$, pre-change intensity function $\lambda^*$ and post-change intensity $\lambda_a^*$. Suppose that $p \geq q$ and that*

$$\sqrt{\sum_{k=r+1}^\infty \sigma_k^2(\lambda^* - \lambda_a^*)} \leq \frac{\|\lambda^* - \lambda_a^*\|_{\mathbf{L}_2}}{5}. \tag{17}$$

*Let $\kappa = \|\lambda^* - \lambda_a^*\|_{\mathbf{L}_2}$. Suppose in addition that*

$$\Delta \geq C_{\mathrm{lag}} r (\log(\mathfrak{b})/\kappa)^{2+p/\gamma} \tag{18}$$

*where $C$ is a sufficiently large constant only depending on $C_\alpha$ in (16). Then*

$$\mathbb{P}\left( \|\widehat{A}_{\Delta+\mathfrak{b},\mathfrak{b}}[r]\|_{\mathrm{F}} \geq \tau_{\Delta+\mathfrak{b},\mathfrak{b}} \right) \geq 1 - \mathfrak{b}^{-3},$$

*where $\widehat{A}_{\Delta+\mathfrak{b},\mathfrak{b}}[r]$ is defined according to (15).*

*Proof.* We apply Lemma B.4 to $\widehat{A}_{\Delta+\mathfrak{b},\mathfrak{b}}[r]$ with the difference of ground truth intensity functions being $\delta^* = \lambda^* - \lambda_a^*$ to deduce that, with probability at least $1 - 2\epsilon\mathfrak{b}^{-3}$,

$$\left| \|\widehat{A}_{\Delta+\mathfrak{b},\mathfrak{b}}[r]\|_{\mathrm{F}} - \|\delta^*\|_{\mathbf{L}_2} \right|$$

$$\leq C_2 m_{\Delta+\mathfrak{b},\mathfrak{b}}^{-\gamma} + C_\epsilon \sqrt{r} \left\{ \log(\mathfrak{b}) + \log(m_{\Delta+\mathfrak{b},\mathfrak{b}}) \right\} \sqrt{\frac{m_{\Delta+\mathfrak{b},\mathfrak{b}}^p}{\Delta}} + 4\sqrt{\sum_{k=r+1}^\infty \sigma_k^2(\delta^*)},$$

$$\leq C_2 m_{\Delta+\mathfrak{b},\mathfrak{b}}^{-\gamma} + 2C_\epsilon \sqrt{r} \log(\mathfrak{b}) \sqrt{\frac{m_{\Delta+\mathfrak{b},\mathfrak{b}}^p}{\Delta}} + 4\sqrt{\sum_{k=r+1}^\infty \sigma_k^2(\delta^*)} \tag{19}$$

Here the second inequality follows from assumption that $N_{\mathrm{train}}$ is sufficiently large, $\mathfrak{b} \geq N_{\mathrm{train}}$, and the observation that

$$m_{\Delta+\mathfrak{b},\mathfrak{b}} = \lceil (\Delta/r)^{1/(2\gamma+p)} \rceil \leq \mathfrak{b}.$$

It follows that with probability at least $1 - 2\epsilon\mathfrak{b}^{-3}$,

$$\|\widehat{A}_{\Delta+\mathfrak{b},\mathfrak{b}}[r]\|_{\mathrm{F}} \geq \frac{\|\delta^*\|_{\mathbf{L}_2}}{5} - \left( C_2 m_{\Delta+\mathfrak{b},\mathfrak{b}}^{-\gamma} + 2C_\epsilon \sqrt{r} \log(\mathfrak{b}) \sqrt{\frac{m_{\Delta+\mathfrak{b},\mathfrak{b}}^p}{\Delta}} \right)$$

$$= \frac{\kappa}{5} - \left( C_2 m_{\Delta+\mathfrak{b},\mathfrak{b}}^{-\gamma} + 2C_\epsilon \sqrt{r} \log(\mathfrak{b}) \sqrt{\frac{m_{\Delta+\mathfrak{b},\mathfrak{b}}^p}{\Delta}} \right)$$

$$\geq \frac{\kappa}{5} - C_3 \log(\mathfrak{b})(r/\Delta)^{\gamma/(2\gamma+p)} > \frac{\kappa}{6}, \tag{20}$$

where the first inequality follows from (19) and (17), the second inequality follows from $m_{\Delta+\mathfrak{b},\mathfrak{b}} = \lceil (\Delta/r)^{1/(2\gamma+p)} \rceil$, and the last inequality follows from (18) with sufficiently large constant $C_{\mathrm{lag}}$. Note that

$$\tau_{\Delta+\mathfrak{b},\mathfrak{b}} = C_\alpha \left(\frac{r}{\Delta}\right)^{\gamma/(2\gamma+p)} \log(\Delta + \mathfrak{b}) \leq 2C_\alpha \left(\frac{r}{\Delta}\right)^{\gamma/(2\gamma+p)} \log(\mathfrak{b}) \leq \kappa/6. \tag{21}$$

Here the equality follows from (16), the first inequality follows from the fact that $\Delta \leq \mathfrak{b}$, and the second inequality follows from (18) with sufficiently large $C_{\mathrm{lag}}$. The desired result follows from (20) and (21). $\qquad\square$

## B. Deviation Bounds

**Theorem B.1** (Singular value decomposition in function space). *Let $F(y, z) : \mathbb{R}^p \times \mathbb{R}^q \to \mathbb{R}$ be any function such that $\|F\|_{\mathbf{L}_2(\mathbb{R}^{p+q})} < \infty$. There exists a collection of singular values $\sigma_1(F) \geq \sigma_2(F) \geq \cdots \geq 0$, and two collections of orthonormal basis functions $\{f_\rho(y)\}_{\rho=1}^\infty \subset \mathbf{L}_2(\mathbb{R}^p)$ and $\{g_\rho(z)\}_{\rho=1}^\infty \subset \mathbf{L}_2(\mathbb{R}^q)$ such that*

$$F(y, z) = \sum_{\rho=1}^\infty \sigma_\rho(F) f_\rho(y) g_\rho(z). \tag{22}$$

*Proof.* See Section 6 of (Brézis, 2011). □

Let $\delta^*(x) = \lambda^*(x) - \lambda_a^*(x) : \mathbb{R}^d \to \mathbb{R}$. Suppose the SVD of $\delta^*$ satisfies

$$\delta^*(x) = \delta^*(y, z) = \sum_{\rho=1}^\infty \sigma_\rho(\delta^*) f_\rho^*(y) g_\rho^*(z).$$

For a positive integer $r$, let

$$\delta^*[r](y, z) = \sum_{\rho=1}^r \sigma_\rho(\delta^*) f_\rho^*(y) g_\rho^*(z).$$

Therefore $\delta^*[r]$ is the best rank-$r$ estimate of $\delta^*$. We can represent $\delta^*$ as a matrix in the following way. Let $A^* \in \mathbb{R}^{m^p \times m^q}$ be the coefficient matrix whose $(\mu, \eta)$ entry is

$$A_{(\mu,\eta)}^* = \iint_{\mathbb{R}^{p+q}} \delta^*(y, z) \Phi_\mu(y) \Psi_\eta(z) dy dz. \tag{23}$$

We approximate $\delta^*(y, z)$ by

$$\delta_m^*(y, z) = \sum_{\mu=1}^{m^p} \sum_{\eta=1}^{m^q} A_{(\mu,\eta)}^* \Phi_\mu(y) \Psi_\eta(z). \tag{24}$$

It was shown in Appendix G1 of (Peng et al., 2024) that if $\{\phi\}_{k=1}^\infty$ are chosen to be the univariate Legendre polynomial bases, then

$$\|\delta^* - \delta_m^*\|_{\mathbf{L}_2} \leq C \|\delta^*\|_{W^{2,\gamma}} \cdot m^{-\gamma} \tag{25}$$

where $\|\delta^*\|_{W^{2,\gamma}}$ is the Sobolev norm of $\delta^*$.

**Definition B.2.** Let $N_1, N_2 \in \mathbb{Z}_+$ be such that $N_1 + N_2 \leq N$. Let $\{X^{(i)}\}_{i=1}^{N_1+N_2}$ be a collection of PPPs satisfying Definition 1.1. Suppose that the intensity function of $\{X^{(i)}\}_{i=1}^{N_1}$ is

$$\lambda^*(x) : \mathbb{R}^d \to \mathbb{R}^+,$$

and the intensity function of $\{X^{(i)}\}_{i=N_1+1}^{N_1+N_2}$ is

$$\lambda_a^*(x) : \mathbb{R}^d \to \mathbb{R}^+.$$

Let $\widehat{A}^{(i)} \in \mathbb{R}^{m^p \times m^q}$ be such that

$$\widehat{A}_{(\mu,\eta)}^{(i)} = \sum_{x=(y,z) \in X^{(i)}} \Phi_\mu(y) \Psi_\eta(z).$$

Define

$$\widehat{A} = \frac{1}{N_1} \sum_{i=1}^{N_1} \widehat{A}^{(i)} - \frac{1}{N_2} \sum_{i=N_1+1}^{N_1+N_2} \widehat{A}^{(i)}. \tag{26}$$

With $x = (x_1, \ldots, x_d)$, $y = (x_1, \ldots, x_p)$ and $z = (x_{p+1}, \ldots, x_{p+q})$, we can write

$$\widehat{\delta}(y, z) = \sum_{\mu=1}^{m^p} \sum_{\eta=1}^{m^q} \widehat{A}_{(\mu,\eta)} \Phi_\mu(y) \Phi_\eta(z). \tag{27}$$

**Lemma B.3.** *Let $\widehat{\delta}$ be defined in* (27). *Then*

$$\mathbb{E}\widehat{\delta} = \delta_m^*.$$

*Proof.* Note that by Theorem C.4, when $i \leq N_1$,

$$\mathbb{E}(\widehat{A}_{(\mu,\eta)}^{(i)}) = \iint_{\mathbb{R}^{p+q}} \Phi_\mu(y)\Psi_\eta(z)\lambda^*(y,z)dydz.$$

Therefore

$$\mathbb{E}(\widehat{A}_{(\mu,\eta)}) = \iint_{\mathbb{R}^{p+q}} \Phi_\mu(y)\Psi_\eta(z)(\lambda^*(y,z) - \lambda_a^*(y,z))dydz = A_{\mu,\eta}^*, \tag{28}$$

where the last equality follows from (23). Therefore

$$\mathbb{E}(\widehat{\delta}(y,z)) = \sum_{\mu=1}^{m^p}\sum_{\eta=1}^{m^q} \mathbb{E}(\widehat{A}_{(\mu,\eta)})\Phi_\mu(y)\Phi_\eta(z) = \sum_{\mu=1}^{m^p}\sum_{\eta=1}^{m^q} A_{(\mu,\eta)}^*\Phi_\mu(y)\Phi_\eta(z) = \delta_m^*(y,z).$$

$\square$

**Lemma B.4.** *Let $\widehat{A} \in \mathbb{R}^{m^p \times m^q}$ be defined in* (26) *and*

$$\widehat{A}[r] = \text{SVD}(\widehat{A}, r).$$

*Suppose in that there exists an absolute constant $C_1$ such that*

$$\max\{\|\lambda^*\|_\infty, \|\lambda_a^*\|_\infty\} < C_1 \quad \max\{\|\lambda^*\|_{W^{2,\gamma}}, \|\lambda_a^*\|_{W^{2,\gamma}}\} < C_1,$$

*and that*

$$N_1 \geq N_2 \geq m^{\max\{p,q\}}. \tag{29}$$

*Then for any $\epsilon > 0$, with probability at least $1 - 2\epsilon N^{-3}$, it holds that*

$$\left| \|\widehat{A}[r]\|_F - \|\delta^*\|_{\mathbf{L}_2} \right| \leq C_2 m^{-\gamma} + C_\epsilon \sqrt{r}\{\log^2(N) + \log^2(m)\}\sqrt{\frac{m^{\max\{p,q\}}}{N_2}} + 4\sqrt{\sum_{k=r+1}^{\infty} \sigma_k^2(\delta^*)},$$

*where $C_2 > 0$ is an absolute constant depending only on $C_1$, and $C_\epsilon > 0$ is an absolute constant depending only on $C_1$ and $\epsilon$.*

*Proof.* Let

$$\widehat{\delta}[r](y,z) = \sum_{\mu=1}^{m^p}\sum_{\eta=1}^{m^q} \widehat{A}[r]_{(\mu,\eta)}\Phi_\mu(y)\Phi_\eta(z).$$

Observe that

$$\|\delta^* - \widehat{\delta}[r]\|_{\mathbf{L}_2} \leq \|\delta^* - \delta_m^*\|_{\mathbf{L}_2} + \|\delta_m^* - \widehat{\delta}[r]\|_{\mathbf{L}_2} = \|\delta^* - \delta_m^*\|_{\mathbf{L}_2} + \|A^* - \widehat{A}[r]\|_F, \tag{30}$$

where $\delta_m^*$ is defined in (24), and the equality follows from Lemma C.2. In addition, by Theorem C.3,

$$\|A^* - \widehat{A}[r]\|_F \leq 4\sqrt{\sum_{k=r+1}^{\infty} \sigma_k^2(A^*)} + 4\sqrt{r}\|A^* - \widehat{A}\|.$$

Note that

$$\sqrt{\sum_{k=r+1}^{\infty} \sigma_k^2(A^*)} = \sqrt{\sum_{k=r+1}^{\infty} \sigma_k^2(\delta_m^*)} \leq \sqrt{\sum_{k=r+1}^{\infty} \sigma_k^2(\delta_m^* - \delta^*)} + \sqrt{\sum_{k=r+1}^{\infty} \sigma_k^2(\delta^*)}$$

$$\leq \|\delta_m^* - \delta^*\|_{\mathrm{F}} + \sqrt{\sum_{k=r+1}^{\infty} \sigma_k^2(\delta^*)},$$

where the equality follows from Lemma C.2, the first inequality follows from Lemma C.1, and the second inequality follows from the fact that

$$\|\delta_m^* - \delta^*\|_{\mathrm{F}} = \sqrt{\sum_{k=1}^{\infty} \sigma_k^2(\delta_m^* - \delta^*)} \geq \sqrt{\sum_{k=r+1}^{\infty} \sigma_k^2(\delta_m^* - \delta^*)}.$$

Therefore

$$\|A^* - \widehat{A}[r]\|_{\mathrm{F}} \leq 4\|\delta_m^* - \delta^*\|_{\mathrm{F}} + 4\sqrt{\sum_{k=r+1}^{\infty} \sigma_k^2(\delta^*)} + 4\sqrt{r}\|A^* - \widehat{A}\|. \tag{31}$$

Combining (30) and (31), we have

$$\|\delta^* - \widehat{\delta}[r]\|_{\mathbf{L}_2} \leq 5\|\delta_m^* - \delta^*\|_{\mathrm{F}} + 4\sqrt{\sum_{k=r+1}^{\infty} \sigma_k^2(\delta^*)} + 4\sqrt{r}\|A^* - \widehat{A}\|$$

$$\leq C_2 m^{-\gamma} + 4\sqrt{\sum_{k=r+1}^{\infty} \sigma_k^2(\delta^*)} + 4\sqrt{r}\|A^* - \widehat{A}\|,$$

where the last inequality follows from (25). By Lemma B.5 and the fact that $A^* = \mathbb{E}(\widehat{A})$ as specified in (28), we have with probability at least $1 - 2\epsilon N^{-3}$

$$\|\widehat{A} - \mathbb{E}(\widehat{A})\| \leq C_\epsilon' \sqrt{\frac{\max\{\|\lambda^*\|_\infty, \|\lambda_a^*\|_\infty\}(m^p + m^q)}{N_1 \wedge N_2}} \log^2(N \vee (m^p + m^q)).$$

Therefore with probability at least $1 - 2N^{-3}$, it holds that

$$\left| \|\widehat{\delta}[r]\|_{\mathbf{L}_2} - \|\delta^*\|_{\mathbf{L}_2} \right| \leq \|\delta^* - \widehat{\delta}[r]\|_{\mathbf{L}_2}$$

$$\leq C_2 m^{-\gamma} + C_\epsilon' \sqrt{r} \sqrt{\frac{\max\{\|\lambda^*\|_\infty, \|\lambda_a^*\|_\infty\}(m^p + m^q)}{N_1 \wedge N_2}} \log^2(N \vee (m^p + m^q)) + 4\sqrt{\sum_{k=r+1}^{\infty} \sigma_k^2(\delta^*)}$$

$$\leq C_2 m^{-\gamma} + C_\epsilon \sqrt{r} \{\log^2(N) + \log^2(m)\} \left(\sqrt{\frac{m^p + m^q}{N_2}}\right) + 4\sqrt{\sum_{k=r+1}^{\infty} \sigma_k^2(\delta^*)},$$

where the last inequality follows from (29). The desired result follows from the fact that $\|\widehat{\delta}[r]\|_{\mathbf{L}_2} = \|\widehat{A}[r]\|_{\mathrm{F}}$. □

## B.1. Poisson Matrix Properties

**Lemma B.5.** *Let $\widehat{A}$ be defined in* (26). *with probability at least $1 - 2N^{-3}$*

$$\|\widehat{A} - \mathbb{E}(\widehat{A})\| \leq C\sqrt{\frac{\max\{\|\lambda^*\|_\infty, \|\lambda_a^*\|_\infty\}(m^p + m^q)}{N_1 \wedge N_2}} \log^2(N \vee (m^p + m^q)).$$

*Proof.* Under Definition B.2, we can apply Theorem C.9, with $Y_i = \widehat{A}^{(i)}$, $F_{\mu,\eta}(y,z) = \Phi_\mu(y)\Psi_\eta(z)$, $1 \le \mu \le m^p$ and $1 \le \mu \le m^q$. We have with probability at least $1 - 2N^{-3}$,

$$\Big\| \sum_{i=1}^{N_1} \widehat{A}^{(i)} - \mathbb{E}(\sum_{i=1}^{N_1} \widehat{A}^{(i)}) \Big\| \le C\sqrt{N_1}\big(\sqrt{\nu_{\max}} + L\big) \log^2(N \vee (m^p + m^q)),$$

and

$$\Big\| \sum_{i=N_1+1}^{N_1+N_2} \widehat{A}^{(i)} - \mathbb{E}(\sum_{i=N_1+1}^{N_1+N_2} \widehat{A}^{(i)}) \Big\| \le C\sqrt{N_2}\big(\sqrt{\nu_{\max}} + L\big) \log^2(N \vee (m^p + m^q)),$$

where

$$\nu_i = \max\Big\{ \Big\| \int_{\mathbb{X}} F(x)F(x)^\top \lambda_i^*(x)\,dx \Big\|_{\mathrm{op}}, \ \Big\| \int_{\mathbb{X}} F(x)^\top F(x)\lambda_i^*(x)\,dx \Big\|_{\mathrm{op}} \Big\}, \qquad \nu_{\max} = \max_{1 \le i \le n} \nu_i.$$

By the orthogonality of $\Phi$ and $\Psi$, the proof of Lemma 5 in (Xu et al., 2025) gives that

$$\nu_{\max} \le \max\{\|\lambda^*\|_\infty, \|\lambda_a^*\|_\infty\}(m^p + m^q).$$

Therefore, we have with probability at least $1 - 2N^{-3}$

$$\|\widehat{A} - \mathbb{E}(\widehat{A})\| \le \frac{1}{N_1}\|\sum_{i=1}^{N_1} \widehat{A}^{(i)} - \mathbb{E}(\sum_{i=1}^{N_1} \widehat{A}^{(i)})\| + \frac{1}{N_2}\|\sum_{i=N_1+1}^{N_1+N_2} \widehat{A}^{(i)} - \mathbb{E}(\sum_{i=N_1+1}^{N_1+N_2} \widehat{A}^{(i)})\|$$

$$\le C\sqrt{\frac{\max\{\|\lambda^*\|_\infty, \|\lambda_a^*\|_\infty\}(m^p + m^q)}{N_1 \wedge N_2}} \log^2(N \vee (m^p + m^q)).$$

$\square$

## C. Auxiliary Results

**Lemma C.1** (Lemma 25 in Xu et al. 2025: Mirsky in Hilbert space)**.** *Suppose $A$ and $B$ are two compact operators in $\mathcal{W} \otimes \mathcal{W}'$, where $\mathcal{W}$ and $\mathcal{W}'$ are two separable Hilbert spaces. Let $\{\sigma_k(A)\}_{k=1}^\infty$ be the singular values of $\mathcal{A}$ in the decreasing order, and $\{\sigma_k(\mathcal{B})\}_{k=1}^\infty$ be the singular values of $\mathcal{B}$ in the decreasing order. Then*

$$\sum_{k=1}^\infty (\sigma_k(A) - \sigma_k(B))^2 \le \|A - B\|_{\mathrm{F}}^2 = \sum_{k=1}^\infty \sigma_k^2(A - B).$$

**Lemma C.2.** *For any $A \in \mathbb{R}^{m^p \times m^q}$, let $\mathcal{F}$ be a map from $\mathbb{R}^{m^p \times m^q}$ to $\mathbf{L}_2(\mathbb{R}^p \times \mathbb{R}^q)$ such that*

$$\mathcal{F}(A)(y,z) = \sum_{\mu=1}^{m^p} \sum_{\eta=1}^{m^q} A_{\mu,\eta} \Phi_\mu(y) \Psi_\eta(z), \tag{32}$$

*where $\{\Phi_\mu(y)\}_{\mu=1}^{m^p}$ and $\{\Psi_\mu(z)\}_{\eta=1}^{m^q}$ are generic basis functions of $\mathbf{L}_2(\mathbb{R}^p)$ and $\mathbf{L}_2(\mathbb{R}^q)$ respectively. Then*

$$\sigma_k(A) = \sigma_k(\mathcal{F}(A)) \quad \text{for any } k \in \mathbb{Z}^+.$$

*Proof.* Note that the map $\mathcal{F}$ is distance preserving in the sense that for any $A, B \in \mathbb{R}^{m^p \times m^q}$,

$$\|A - B\|_{\mathrm{F}} = \|\mathcal{F}(A) - \mathcal{F}(B)\|_{\mathbf{L}_2}.$$

Since $\mathcal{F}$ is distance preserving, $\mathcal{F}$ also preserves the singular values. $\square$

**Theorem C.3.** *Let $X$ and $Z$ be two generic matrices in $\mathbb{R}^{p \times q}$ and that*

$$Y = X + Z.$$

*Denote $Y[r] = \mathrm{SVD}(Y, r)$. Then*

$$\|Y[r] - X^*\|_{\mathrm{F}} \;\leq\; (2 + \sqrt{2})\Big(\sqrt{\sum_{i=r+1}^{\min\{m,n\}} \sigma_i^2(X^*)} + \sqrt{r}\,\|Z\|\Big),$$

*Proof.* The desired result is a direct consequence of Lemma 16 in (Xu et al., 2025). $\qquad\qquad\square$

**Theorem C.4** (Theorem 2.2 in Baddeley 2007: Campbell's Theorem). *Let $X \subset \mathbb{R}^d$ be a Poisson process with the intensity function $\lambda^*$. For all measurable function $f : \mathbb{R}^d \to \mathbb{R}$, it holds that*

$$\mathbb{E}\Big(\sum_{x \in X} f(x)\Big) = \int_{\mathbb{R}^d} f(x)\lambda^*(x)dx.$$

**Lemma C.5.** *Let $\lambda \in \mathbf{L}_2(\Omega^{\otimes p} \times \Omega^{\otimes q})$. For any integer $M \geq 1$, define the finite-dimensional subspaces*

$$\mathcal{U}_M = \mathrm{span}\{\Phi_\mu : \mu = 1, \ldots, M^p\} \subset \mathbf{L}_2(\Omega^{\otimes p}), \qquad \mathcal{V}_M = \mathrm{span}\{\Psi_\eta : \eta = 1, \ldots, M^q\} \subset \mathbf{L}_2(\Omega^{\otimes q}),$$

*and let $\mathcal{P}_{\mathcal{U}_M}$ and $\mathcal{P}_{\mathcal{V}_M}$ be the orthogonal projections onto $\mathcal{U}_M$ and $\mathcal{V}_M$.*

*Let $T_\lambda : \mathbf{L}_2(\Omega^{\otimes q}) \to \mathbf{L}_2(\Omega^{\otimes p})$ be the integral operator*

$$(T_\lambda g)(y) = \int_{\Omega^{\otimes q}} \lambda(y, z)\, g(z)\, dz.$$

*Then $T_\lambda$ is Hilbert–Schmidt (hence compact), and we define $\{\sigma_k(\lambda)\}_{k \geq 1}$ as the singular values of $T_\lambda$ (in nonincreasing order).*

*Define the matrix $\mathcal{M}(\lambda) \in \mathbb{R}^{M^p \times M^q}$ by*

$$\mathcal{M}(\lambda)_{(\mu,\eta)} = \iint_{\Omega^{\otimes p} \times \Omega^{\otimes q}} \lambda(y, z)\, \Phi_\mu(y)\, \Psi_\eta(z)\, dy\, dz.$$

*Let $\{\sigma_k(\mathcal{M}(\lambda))\}_{k \geq 1}$ denote the singular values of the matrix $\mathcal{M}(\lambda)$. Then for every integer $R \geq 1$,*

$$\sum_{k>R} \sigma_k^2(\mathcal{M}(\lambda)) \;\leq\; \sum_{k>R} \sigma_k^2(\lambda).$$

*Proof.* **Step 1: $\mathcal{M}(\lambda)$ represents the compressed operator.** Let

$$\Pi_M = \mathcal{P}_{\mathcal{U}_M}\, T_\lambda\, \mathcal{P}_{\mathcal{V}_M} : \mathbf{L}_2(\Omega^{\otimes q}) \to \mathbf{L}_2(\Omega^{\otimes p}).$$

For $1 \leq \mu \leq M^p$ and $1 \leq \eta \leq M^q$, using $\mathcal{P}_{\mathcal{U}_M}\Phi_\mu = \Phi_\mu$ and $\mathcal{P}_{\mathcal{V}_M}\Psi_\eta = \Psi_\eta$,

$$\begin{aligned}
\langle \Phi_\mu,\, \Pi_M \Psi_\eta \rangle_{\mathbf{L}_2(\Omega^{\otimes p})} &= \langle \Phi_\mu,\, T_\lambda \Psi_\eta \rangle \\
&= \int_{\Omega^{\otimes p}} \Phi_\mu(y) \left( \int_{\Omega^{\otimes q}} \lambda(y, z)\Psi_\eta(z)\, dz \right) dy \\
&= \iint_{\Omega^{\otimes p} \times \Omega^{\otimes q}} \lambda(y, z)\Phi_\mu(y)\Psi_\eta(z)\, dy\, dz \\
&= \mathcal{M}(\lambda)_{(\mu,\eta)}.
\end{aligned}$$

Hence, the restriction $\Pi_M|_{\mathcal{V}_M} : \mathcal{V}_M \to \mathcal{U}_M$ has matrix $\mathcal{M}(\lambda)$ in the orthonormal bases $\{\Psi_\eta\}$ and $\{\Phi_\mu\}$. Therefore the nonzero singular values of $\Pi_M$ coincide with those of $\mathcal{M}(\lambda)$, and

$$\sum_{k>R} \sigma_k^2(\mathcal{M}(\lambda)) = \inf_{\mathrm{rank}(A) \leq R} \|\Pi_M - A\|_{\mathrm{HS}}^2,$$

where $\|\cdot\|_{\text{HS}}$ is the Hilbert–Schmidt norm.

**Step 2: Eckart–Young for Hilbert–Schmidt operators.** Since $T_\lambda$ is Hilbert–Schmidt, it admits an SVD $T_\lambda = \sum_{k\geq 1} \sigma_k(\lambda)\, u_k \otimes v_k$ with orthonormal $\{u_k\}, \{v_k\}$. Let $T_{\lambda,R} = \sum_{k=1}^{R} \sigma_k(\lambda)\, u_k \otimes v_k$. Then $T_{\lambda,R}$ has rank at most $R$ and

$$\inf_{\text{rank}(B)\leq R} \|T_\lambda - B\|_{\text{HS}}^2 = \|T_\lambda - T_{\lambda,R}\|_{\text{HS}}^2 = \sum_{k>R} \sigma_k^2(\lambda).$$

**Step 3: Compression by orthogonal projections is a contraction in HS norm.** For any Hilbert–Schmidt operator $A$ and orthogonal projections $P, Q$,

$$\|PAQ\|_{\text{HS}}^2 = \text{tr}\big((PAQ)^*(PAQ)\big) = \text{tr}\big(QA^*PAQ\big) \leq \text{tr}\big(QA^*AQ\big) \leq \text{tr}(A^*A) = \|A\|_{\text{HS}}^2,$$

where we used $0 \leq P \leq I$ and $0 \leq Q \leq I$ and the monotonicity of the trace on positive trace-class operators.

**Step 4: Compare best rank-$R$ errors.** Let $T_{\lambda,R}$ be defined in **Step 2**. Then $\mathcal{P}_{\mathcal{U}_M} T_{\lambda,R} \mathcal{P}_{\mathcal{V}_M}$ has rank at most $R$. By optimality of the best rank-$R$ approximation of $\Pi_M$ and **Step 3**,

$$\sum_{k>R} \sigma_k^2\big(\mathcal{M}(\lambda)\big) = \inf_{\text{rank}(A)\leq R} \|\Pi_M - A\|_{\text{HS}}^2$$

$$\leq \|\Pi_M - \mathcal{P}_{\mathcal{U}_M} T_{\lambda,R} \mathcal{P}_{\mathcal{V}_M}\|_{\text{HS}}^2$$

$$= \|\mathcal{P}_{\mathcal{U}_M}(T_\lambda - T_{\lambda,R})\mathcal{P}_{\mathcal{V}_M}\|_{\text{HS}}^2$$

$$\leq \|T_\lambda - T_{\lambda,R}\|_{\text{HS}}^2 = \sum_{k>R} \sigma_k^2(\lambda).$$

This proves the claimed inequality. $\square$

**Lemma C.6.** *Suppose $\{\sigma_k(F)\}_{k=1}^{\infty}$ are the singular values the function $F$ and that $\{\sigma_k(F)\}_{k=1}^{\infty}$ decays at a polynomial rate. Then there exists a constant $r$ depending only on the decay rate of $\{\sigma_k(F)\}_{k=1}^{\infty}$ such that*

$$\sqrt{\sum_{k=r+1}^{\infty} \sigma_k^2(F)} \;\leq\; \frac{\|F\|_{\mathbf{L}_2}}{5}.$$

*Proof.* Without lost of generality, suppose that $\sigma_k(F) = ck^{-a}$ and $\|F\|_{\mathbf{L}_2} = 1$. Since

$$\sum_{k=r+1}^{\infty} \sigma_k^2(F) = \sum_{k=1}^{\infty} c^2 k^{-2a} = \|F\|_{\mathbf{L}_2}^2 = 1,$$

it follows that $a > 1/2$. Since $a > 1/2$,

$$\lim_{r\to\infty} \sum_{k=r+1}^{\infty} c^2 k^{-2a} = 0.$$

Therefore there exists $r \in \mathbb{Z}^+$ depending only on $a$ and $c$ such that $\sqrt{\sum_{k=r+1}^{\infty} \sigma_k^2(F)} \leq \frac{\|F\|_{\mathbf{L}_2}}{5}$ $\square$

**Theorem C.7** (Lemma 5 in Xu et al. 2025: Matrix Bernstein inequality for a Poisson point process). *Let $X$ be an inhomogeneous Poisson point process on a compact set $\mathbb{X} \subset \mathbb{R}^d$ with intensity $\lambda : \mathbb{X} \to \mathbb{R}_+$ satisfying $\|\lambda\|_\infty < \infty$. Let $F : \mathbb{X} \to \mathbb{R}^{d_1 \times d_2}$ be measurable and assume $\sup_{x\in\mathbb{X}} \|F(x)\|_{\text{op}} \leq L < \infty$. Define the matrix variance proxy*

$$\nu = \max\left\{ \left\|\int_{\mathbb{X}} F(x)F(x)^\top \lambda(x)\, dx\right\|_{\text{op}}, \left\|\int_{\mathbb{X}} F(x)^\top F(x)\lambda(x)\, dx\right\|_{\text{op}} \right\}.$$

*Then for all $t \geq 0$,*

$$\mathbb{P}\left( \left\|\sum_{x\in X} F(x) - \int_{\mathbb{X}} F(x)\lambda(x)\, dx\right\|_{\text{op}} \geq t \right) \leq (d_1 + d_2)\exp\left( -\frac{t^2/2}{\nu + Lt/3} \right).$$

*Equivalently, for any $a \geq 2$, with probability at least $1 - 2(\max\{d_1, d_2\})^{1-a}$,*

$$\left\| \sum_{x \in N} F(x) \ - \ \int_{\mathbb{X}} F(x) \lambda(x) \, dx \right\|_{\text{op}} \leq \sqrt{2a\, \nu \, \log(d_1 + d_2)} + \frac{2a}{3} L \log(d_1 + d_2).$$

**Theorem C.8** (Theorem 1 in Banna et al. 2016: Matrix Bernstein inequality for geometrically $\beta$-mixing self-adjoint matrices)**.** *Let $\{X_i\}_{i=1}^n$ be a sequence of centered, self-adjoint random matrices in $\mathbb{R}^{d \times d}$ with $\mathbb{E}[X_i] = 0$ and satisfying*

$$\lambda_{\max}(X_i) \leq M \qquad \text{a.s. for all } i = 1, \ldots, n.$$

*Let $\beta(k)$ be the $\beta$-mixing coefficients of the sequence $\{X_i\}$, and assume geometric $\beta$-mixing: there exists $c > 0$ such that*

$$\beta(k) \leq e^{-c(k-1)} \qquad \text{for all } k \geq 1.$$

*Define*

$$v^2 = \sup_{\emptyset \neq K \subset \{1, \ldots, n\}} \frac{1}{|K|} \lambda_{\max}\left( \mathbb{E}\Big( \sum_{i \in K} X_i \Big)^2 \right),$$

*and*

$$\iota(c, n) = \frac{\log n}{\log 2} \max\left\{ 2, \frac{32 \log n}{c \log 2} \right\}.$$

*Then for all $x > 0$ and all integers $n > 2$,*

$$\mathbb{P}\left( \lambda_{\max}\left( \sum_{i=1}^n X_i \right) \geq x \right) \leq d \exp\left( -\frac{Cx^2}{v^2 n + c^{-1}M^2 + xM\,\iota(c, n)} \right),$$

*where $C > 0$ is a universal constant.*

**Theorem C.9** (Matrix Bernstein for geometrically $\beta$-mixing Poisson point process windows)**.** *Let $\{X^{(i)}\}_{i=1}^n$ be a sequence of (possibly nonstationary) Poisson point processes on a compact set $\mathbb{X} \subset \mathbb{R}^d$ with intensity functions $\lambda_i : \mathbb{X} \to \mathbb{R}_+$. Let $F : \mathbb{X} \to \mathbb{R}^{d_1 \times d_2}$ be measurable and continuous, and assume*

$$\sup_{x \in \mathbb{X}} \|F(x)\|_{\text{op}} \leq L < \infty.$$

*For each $i$, define the centered rectangular matrix*

$$Y_i = \sum_{x \in X^{(i)}} F(x) \ - \ \int_{\mathbb{X}} F(x) \lambda_i(x) \, dx \in \mathbb{R}^{d_1 \times d_2},$$

*and define the (rectangular) variance proxy*

$$\nu_i = \max\left\{ \Big\| \int_{\mathbb{X}} F(x) F(x)^\top \lambda_i(x) \, dx \Big\|_{\text{op}}, \ \Big\| \int_{\mathbb{X}} F(x)^\top F(x) \lambda_i(x) \, dx \Big\|_{\text{op}} \right\}, \qquad \nu_{\max} = \max_{1 \leq i \leq n} \nu_i.$$

*Assume the dependence across windows satisfies the $\beta$-mixing condition: the sequence $\{N^{(i)}\}_{i \geq 1}$ is $\beta$-mixing with coefficients $\{\beta(k)\}_{k \geq 1}$ such that for some constants $c > 0$,*

$$\beta(k) \leq e^{-c(k-1)}, \qquad k \geq 1.$$

*Then, with probability at least $1 - \delta$,*

$$\Big\| \sum_{i=1}^n Y_i \Big\|_{\text{op}} \ \leq \ C\sqrt{n}\big(\sqrt{\nu_{\max}} + L\big) \log^2\Big( \frac{n(d_1 + d_2)}{\delta} \Big), \tag{33}$$

*where $C > 0$ is a universal constant.*

*Proof of Theorem C.9.* **Step 1.** For each $i$, define the Hermitian dilation

$$\overline{Y}_i = \begin{pmatrix} 0 & Y_i \\ Y_i^\top & 0 \end{pmatrix}.$$

Then $\overline{Y}_i$ is self-adjoint and

$$\|\overline{Y}_i\|_{\mathrm{op}} = \|Y_i\|_{\mathrm{op}}, \qquad \Big\|\sum_{i=1}^n \overline{Y}_i\Big\|_{\mathrm{op}} = \Big\|\overline{\sum_{i=1}^n Y_i}\Big\|_{\mathrm{op}} = \Big\|\sum_{i=1}^n Y_i\Big\|_{\mathrm{op}}.$$

Hence it suffices to control $\|\sum_{i=1}^n \overline{Y}_i\|_{\mathrm{op}}$. We consider the following decomposition

$$\overline{Y}_i = Z_i + \overline{Y}_i \mathbf{1}\{\|\overline{Y}_i\|_{\mathrm{op}} > U\} = Z_i^0 + \mathbb{E}Z_i + \overline{Y}_i \mathbf{1}\{\|\overline{Y}_i\|_{\mathrm{op}} > U\},$$

where, for a fixed $U > 0$, $Z_i = \overline{Y}_i \mathbf{1}\{\|\overline{Y}_i\|_{\mathrm{op}} \le U\}$ and $Z_i^0 = Z_i - \mathbb{E}Z_i$.

**Step 2.** For all $i$,
$$\lambda_{\max}(Z_i^0) \le \|Z_i^0\|_{\mathrm{op}} \le \|Z_i\|_{\mathrm{op}} + \|\mathbb{E}Z_i\|_{\mathrm{op}} \le U + U = 2U \quad \text{a.s.}$$

Also, $\mathbb{E}Z_i^0 = 0$ by construction. Moreover, each $Z_i^0$ is a measurable function of $N^{(i)}$ only. By Definition 1.1, for any $k \ge 1$,

$$\beta_{Z_i^0}(k) \le \beta(k) \le e^{-c(k-1)}.$$

We apply Theorem C.8 to $\{Z_i^0\}_{i\in\mathbb{Z}}$ with $M = 2U$. Note that

$$\Big\|\sum_{i=1}^n Z_i^0\Big\|_{\mathrm{op}} = \max\Big\{\lambda_{\max}\Big(\sum_{i=1}^n Z_i^0\Big), \ \lambda_{\max}\Big(-\sum_{i=1}^n Z_i^0\Big)\Big\}.$$

Applying the same bound to $(-Z_i^0)$ and a union bound yields

$$\mathbb{P}\Big(\Big\|\sum_{i=1}^n Z_i^0\Big\|_{\mathrm{op}} \ge x\Big) \le 2(d_1 + d_2)\exp\Big(-\frac{Cx^2}{v_U^2 n + 4c^{-1}U^2 + 2Ux\iota(c,n)}\Big).$$

Equivalently, we have with probability at least $1 - \delta/2$,

$$\Big\|\sum_{i=1}^n Z_i^0\Big\|_{\mathrm{op}} \le C_0\left(\sqrt{\Big(nv_U^2 + 4c^{-1}U^2\Big)\log\frac{4(d_1+d_2)}{\delta}} + 2U\iota(c,n)\log\frac{4(d_1+d_2)}{\delta}\right).$$

**Step 3.** Summing over $i$ gives

$$\sum_{i=1}^n \overline{Y}_i = \sum_{i=1}^n Z_i^0 + \sum_{i=1}^n \mathbb{E}Z_i + \sum_{i=1}^n \overline{Y}_i \mathbf{1}\{\|\overline{Y}_i\|_{\mathrm{op}} > U\}.$$

We denote the truncation bias by
$$b(U) = \max_{i=1}^n \Big\|\mathbb{E}\big[\overline{Y}_i \mathbf{1}\{\|\overline{Y}_i\|_{\mathrm{op}} > U\}\big]\Big\|_{\mathrm{op}}.$$

Note that $\mathbb{E}\overline{Y}_i = 0$ by Theorem C.4, since each $N^{(i)}$ is a PPP with intensity $\lambda_i$. Hence $\mathbb{E}Z_i = -\mathbb{E}\big[\overline{Y}_i\mathbf{1}\{\|\overline{Y}_i\|_{\mathrm{op}} > U\}\big]$. Therefore,

$$\Big\|\sum_{i=1}^n \mathbb{E}Z_i\Big\|_{\mathrm{op}} \le \sum_{i=1}^n \Big\|\mathbb{E}\big[\overline{Y}_i\mathbf{1}\{\|\overline{Y}_i\|_{\mathrm{op}} > U\}\big]\Big\|_{\mathrm{op}} \le nb(U).$$

Define the tail event $\mathcal{E}_U = \{\max_{1 \le i \le n} \|\overline{Y}_i\|_{\mathrm{op}} \le U\}$. On $\mathcal{E}_U$, the random tail term vanishes: $\sum_{i=1}^n \overline{Y}_i\mathbf{1}\{\|\overline{Y}_i\|_{\mathrm{op}} > U\} = 0$. Hence on $\mathcal{E}_U$,

$$\Big\|\sum_{i=1}^n \overline{Y}_i\Big\|_{\mathrm{op}} \le \Big\|\sum_{i=1}^n Z_i^0\Big\|_{\mathrm{op}} + nb(U).$$

**Step 4.** Recall that

$$Y_i = \sum_{x \in X^{(i)}} F(x) - \int_{\mathbb{X}} F(x)\lambda_i(x)\,dx \in \mathbb{R}^{d_1 \times d_2}.$$

By Theorem C.7, we have that

$$\mathbb{P}\big(\|\bar{Y}_i\|_{\mathrm{op}} \geq t\big) = \mathbb{P}\big(\|Y_i\|_{\mathrm{op}} \geq t\big) \leq (d_1 + d_2)\exp\Big(-\frac{t^2}{2(\nu_i + Lt/3)}\Big) \leq (d_1 + d_2)\exp\Big(-\frac{t^2}{2(\nu_{\max} + Lt/3)}\Big).$$

For $\mathcal{E}_U$, apply a union bound:

$$\mathbb{P}(\mathcal{E}_U^c) = \mathbb{P}\Big(\max_{1 \leq i \leq n} \|\bar{Y}_i\|_{\mathrm{op}} > U\Big) \leq \sum_{i=1}^n \mathbb{P}(\|\bar{Y}_i\|_{\mathrm{op}} > U) \leq n(d_1 + d_2)\exp\Big(-\frac{U^2}{2(\nu_{\max} + LU/3)}\Big).$$

Set $U_* = \sqrt{2\nu_{\max}u} + 2Lu/3$ and $u = \log\big(2n(d_1 + d_2)/\delta\big)$, we have

$$\frac{U_*^2}{2(\nu_{\max} + LU_*/3)} \geq u, \tag{34}$$

so

$$\mathbb{P}(\mathcal{E}_{U_*}^c) \leq n(d_1 + d_2)\exp(-u) = \frac{\delta}{2}. \tag{35}$$

**Step 5.** Recall that

$$v_{U_*}^2 = \sup_{\varnothing \neq K \subset \{1,\dots,n\}} \frac{1}{|K|}\lambda_{\max}\Big(\mathbb{E}\big(\sum_{i \in K} Z_i^0\big)^2\Big).$$

Fix a nonempty set $K \subset \{1, \dots, n\}$. Expand

$$\mathbb{E}\Big(\sum_{i \in K} Z_i^0\Big)^2 = \sum_{i \in K} \mathbb{E}(Z_i^0)^2 + \sum_{\substack{i,j \in K \\ i \neq j}} \mathbb{E}(Z_i^0 Z_j^0).$$

Thus,

$$\lambda_{\max}\Big(\mathbb{E}\big(\sum_{i \in K} Z_i^0\big)^2\Big) \leq \sum_{i \in K} \big\|\mathbb{E}(Z_i^0)^2\big\|_{\mathrm{op}} + \sum_{\substack{i,j \in K \\ i \neq j}} \big\|\mathbb{E}(Z_i^0 Z_j^0)\big\|_{\mathrm{op}}.$$

*(i) Diagonal terms.* Since $Z_i^0$ is self-adjoint, we have

$$\mathbb{E}(Z_i^0)^2 = \mathbb{E}Z_i^2 - (\mathbb{E}Z_i)^2 \preceq \mathbb{E}(Z_i)^2,$$

and thus

$$\big\|\mathbb{E}(Z_i^0)^2\big\|_{\mathrm{op}} \leq \big\|\mathbb{E}Z_i^2\big\|_{\mathrm{op}} \leq \big\|\mathbb{E}\bar{Y}_i^2\big\|_{\mathrm{op}} = \nu_i \leq \nu_{\max}.$$

*(ii) Off-diagonal terms.* For any $j > i$, by Berbee's coupling lemma for $\beta$-mixing process, there exists $(Z_j^0)'$ such that $(Z_j^0)' \stackrel{d}{=} Z_j^0$, $(Z_j^0)'$ is independent of $Z_i^0$, and $\mathbb{P}((Z_j^0)' \neq Z_j^0) \leq \beta(j - i)$. Since $\|Z_i^0\|_{\mathrm{op}} \leq 2U_*$ a.s., we have

$$\big\|\mathbb{E}(Z_i^0 Z_j^0)\big\|_{\mathrm{op}} = \big\|\mathbb{E}\big[Z_i^0(Z_j^0 - (Z_j^0)')\big]\big\|_{\mathrm{op}} \leq \mathbb{E}\big[\|Z_i^0\|_{\mathrm{op}} \|Z_j^0 - (Z_j^0)'\|_{\mathrm{op}}\big] \leq 8U_*^2\beta(j - i),$$

where the first inequality follows from Jensen's inequality. Therefore,

$$\sum_{i \neq j} \big\|\mathbb{E}(Z_i^0 Z_j^0)\big\|_{\mathrm{op}} \leq 2\sum_{i < j} 8U_*^2\beta(j - i) = 16U_*^2\sum_{h \geq 1}\#\{(i,j) \in K^2 : j - i = h\}\beta(h) \leq 16U_*^2|K|\sum_{h \geq 1}\beta(h).$$

Combining *(i)* and *(ii)*,

$$\lambda_{\max}\Big(\mathbb{E}\big(\sum_{i \in K} X_i\big)^2\Big) \leq |K|\nu_{\max} + 16U_*^2|K|\sum_{h \geq 1}\beta(h) \leq |K|\nu_{\max} + \frac{16U_*^2|K|}{1 - e^{-c}},$$

where the last inequality follows from Definition 1.1. Divide by $|K|$ and take the supremum over $K$:

$$v_{U_*}^2 \leq \nu_{\max} + \frac{16U_*^2}{1 - e^{-c}}.$$

**Step 6.** Recall that

$$b(U_*) = \max_{1 \leq i \leq n} \left\| \mathbb{E}\left[ \bar{Y}_i \, \mathbf{1}\{\|\bar{Y}_i\|_{\mathrm{op}} > U_*\} \right] \right\|_{\mathrm{op}} = \max_{1 \leq i \leq n} \int_{U_*}^{\infty} \mathbb{P}(\|\bar{Y}_i\|_{\mathrm{op}} \geq t) \, dt$$

$$\leq (d_1 + d_2) \int_{U_*}^{\infty} \exp\left( -\frac{t^2}{2(\nu_{\max} + Lt/3)} \right) dt.$$

Before computing the above integral, we simplify the notation by letting $a = \nu_{\max} > 0$, $b = L/3 > 0$ and $\psi(t) = t^2/(2(a + bt))$. A direct computation gives

$$\psi'(t) = \frac{t(2a + bt)}{2(a + bt)^2}, \qquad \psi''(t) = \frac{a^2}{(a + bt)^3} > 0,$$

so $\psi$ is convex and $\psi'$ is nondecreasing. By convexity, for all $t \geq U_*$, $\psi(t) \geq \psi(U_*) + \psi'(U_*)(t - U_*)$. Therefore,

$$\int_{U_*}^{\infty} e^{-\psi(t)} dt \leq e^{-\psi(U_*)} \int_0^{\infty} e^{-\psi'(U_*)s} \, ds = \frac{e^{-\psi(U_*)}}{\psi'(U_*)}.$$

Note that

$$\frac{1}{\psi'(U_*)} = \frac{2(a + bU_*)^2}{U_*(2a + bU_*)} = \frac{2(a + bU_*)}{U_*} \cdot \frac{a + bU_*}{2a + bU_*} \leq \frac{2(a + bU_*)}{U_*},$$

since $2a + bU_* \geq a + bU_*$. We have

$$b(U_*) \leq (d_1 + d_2) \int_{U_*}^{\infty} e^{-\psi(t)} dt \leq (d_1 + d_2) \frac{2(\nu_{\max} + LU_*/3)}{U_*} \exp\left( -\frac{U_*^2}{2(\nu_{\max} + LU_*/3)} \right)$$

$$\leq 2\left( \frac{\nu_{\max}}{U_*} + \frac{L}{3} \right)(d_1 + d_2) \exp(-u)$$

$$\leq \left( \frac{\nu_{\max}}{U_*} + \frac{L}{3} \right) \frac{\delta}{n},$$

where the third inequality follows from (34), the fourth inequality follows from (35). Finally, we have

$$nb(U_*) \leq \delta \left( \sqrt{\frac{\nu_{\max}}{2 \log(2nd/\delta)}} + \frac{L}{3} \right),$$

**Step 7.** Combining the above steps, we have with probability at least $1 - \delta$

$$\left\| \sum_{i=1}^n \bar{Y}_i \right\|_{\mathrm{op}}$$

$$\leq C_0 \left( \sqrt{\left( nv_{U_*}^2 + 4c^{-1}U_*^2 \right) \log \frac{4(d_1 + d_2)}{\delta}} + 2U_* \iota(c, n) \log \frac{4(d_1 + d_2)}{\delta} \right)$$

$$+ \delta \left( \sqrt{\frac{\nu_{\max}}{2 \log(2n(d_1 + d_2)/\delta)}} + \frac{L}{3} \right)$$

$$\leq C_1 \left( \sqrt{n \left( \nu_{\max} \log(2n(d_1 + d_2)/\delta) + L^2 \log^2(2n(d_1 + d_2)/\delta) \right) \log \frac{4d}{\delta}} \right.$$

$$\left. + \left( \sqrt{\nu_{\max} \log(2n(d_1 + d_2)/\delta)} + L \log(2n(d_1 + d_2)/\delta) \right) \iota(c, n) \log \frac{4d}{\delta} \right)$$

$$\leq C_1 \left( \sqrt{\nu_{\max} \log(2n(d_1 + d_2)/\delta)} + L \log(2n(d_1 + d_2)/\delta) \right) \left( \sqrt{n \log \frac{4(d_1 + d_2)}{\delta}} + \iota(c, n) \log \frac{4(d_1 + d_2)}{\delta} \right)$$

$$\leq C_2 \sqrt{n} \left( \sqrt{\nu_{\max}} + L \right) \log^2 \left( \frac{n(d_1 + d_2)}{\delta} \right).$$

$\square$

## D. Online 1D Poisson point processes change detection

For completeness, we introduce a simplified version of Algorithm 3 suitable for detecting change points for Poisson Process time series data in 1D.

Let $\{\phi_\mu\}_{\mu=1}^\infty$ be a collection of orthonormal basis functions of $\mathbf{L}_2(\Omega)$, where $\Omega \subset \mathbb{R}$ is a compact interval. Let $M$ be a positive integer. For each process $X^{(i)} \subset \Omega$, define its intensity matrix by

$$\widehat{\mathcal{V}}^{(i)} \in \mathbb{R}^M, \quad \widehat{\mathcal{V}}_\mu^{(i)} = \sum_{x \in X^{(i)}} \phi_\mu(x). \tag{36}$$

**Theorem D.1.** *Assume $d = 1$ and that the time series $\{X^{(i)}\}_{i \in \mathbb{Z}}$ on a compact domain $\mathbb{X} \subset \mathbb{R}$ satisfies Definition 1.1. Let the univariate orthonormal basis $\{\phi_\mu\}_{\mu \geq 1}$ in (36) be the Legendre polynomials, and suppose the training size $N_{\text{train}}$ is sufficiently large.*

*(a) No change point. Assume (M0) in Definition 1.1 holds with intensity $\lambda^*$ and $\|\lambda^*\|_{W^{2,\gamma}} < \infty$. If the threshold constant $\mathcal{C}_\alpha$ in Algorithm 3 is chosen sufficiently large, then with probability at least $1 - \alpha$, Algorithm 3 never raises an alarm over the entire time horizon.*

*(b) Single change point. Assume (M1) in Definition 1.1 holds with change point $\mathfrak{b} \geq N_{\text{train}}$ and intensities $\lambda^*, \lambda_a^*$ satisfying $\|\lambda^*\|_{W^{2,\gamma}} < \infty$ and $\|\lambda_a^*\|_{W^{2,\gamma}} < \infty$. Let $\kappa = \|\lambda^* - \lambda_a^*\|_{\mathbf{L}_2}$ and define*

$$\Delta = \left\lceil C_{\text{lag}} \left( \log(\mathfrak{b})/\kappa \right)^{2+1/\gamma} \right\rceil, \tag{37}$$

*where $C_{\text{lag}}$ is a sufficiently large constant depending only on $\mathcal{C}_\alpha$. If the window size satisfies $W \geq \Delta$, then with probability at least $1 - \mathfrak{b}^{-3}$, Algorithm 3 raises an alarm within the time interval $(\mathfrak{b}, \mathfrak{b} + \Delta]$.*

*Proof.* The proof of Theorem D.1 is similar and simpler than Theorem 2.4, and will be omitted for brevity. $\square$

## E. Additional Numerical Studies

In this section, we report two additional simulation studies that supplement the main-paper experiments. The first is a 10-dimensional Poisson point process (PPP) experiment in which the intensity functions are *not* exactly low-rank, evaluating the proposed method against three baselines including a neural-network-based detector. The second is a robustness analysis with respect to the rank parameter $r$ and the coordinate partition.

### E.1. 10D Intensity with Non-Low-Rank Difference

We generate a temporally dependent 10D PPP time series on $[0, 1]^{10}$, where the intensity function changes from

$$\lambda^*(x) = z_t^+ \left( 1 + \sin(\pi \sum_{j=1}^{10} x_j) \right) \quad \text{to} \quad \lambda_a^*(x) = z_t^+ \left( 1 + \cos(\pi \sum_{j=1}^{10} x_j) \right)$$

at the change point $\mathfrak{b} = 1400$, with $x \in [0, 1]^{10}$. Temporal dependence is introduced through an autoregressive intensity scale $\{z_t\}$. Note that neither $\lambda^*$ nor $\lambda_a^*$ admits an exact finite-rank representation, so the matricized intensities have an infinite (geometrically decaying) singular-value tail. We use $N_{\text{train}} = 1200$ pre-change samples, $N_{\text{total}} = 1800$ samples in total, and average over 100 Monte Carlo replications.

In addition to the **Matrix**, **MMD**, and **KIE** detectors used in the main paper, we include a fourth, neural-network-based density change-point detector adapted from Gong et al. (2022), which we refer to as the **NN-CUSUM** detector. This detector trains a small permutation-invariant network online to discriminate the current window from pre-change reference windows and monitors the CUSUM of its held-out score; we use the authors' default architecture and optimizer settings, fixed once and held constant across replications. As in the main-paper experiments, all thresholds are calibrated by the block-permutation procedure described in Section 3.1, and tuning parameters of the competing detectors follow their authors' default choices.

---

**Algorithm 3** Online 1D PPP change detection

---

**Input:** Smoothness parameter $\gamma > 0$; window size $W$; threshold constant $\mathcal{C}_\alpha$

▶ **Initialization Stage**
$M \leftarrow \lceil W^{1/(2\gamma+1)} \rceil$
**for** $k = 1$ **to** $W$ **do**
    $L[k] \leftarrow \sum_{i=1}^{N_{\text{train}}-W+k-1} \widehat{\mathcal{V}}^{(i)} \in \mathbb{R}^M$    (computed via (36))
**end for**

▶ **Detection Stage**
$\text{ALARM} \leftarrow \text{FALSE}$
**for** $j = N_{\text{train}} + 1, N_{\text{train}} + 2, \dots$ **do**
    **for** $k = 1$ **to** $W - 1$ **do**
        $L[k] \leftarrow L[k+1]$
    **end for**
    $L[W] \leftarrow L[W] + \widehat{\mathcal{V}}^{(j-1)}$
    **for** $k = 1$ **to** $W$ **do**
        $R[k] \leftarrow \sum_{i=j-W+k}^{j} \widehat{\mathcal{V}}^{(i)}$
    **end for**
    **for** $k = 1$ **to** $W$ **do**
        $n_1 \leftarrow j - W - 1 + k$
        $n_2 \leftarrow W - k + 1$
        $\mathcal{D} \leftarrow n_1^{-1} L[k] - n_2^{-1} R[k]$
        **if** $\|\mathcal{D}\| > \mathcal{C}_\alpha \left(\dfrac{1}{n_2}\right)^{\gamma/(2\gamma+1)} \log(j)$ **then**
            $\text{ALARM} \leftarrow \text{TRUE}$
            **break**
        **end if**
    **end for**
**end for**

---

Table 3 reports the false-alarm rate, correct-detection rate, no-alarm rate, and the average detection delay (ADD) conditional on correct detection. Figure 4 shows the corresponding FAP–ADD trade-off curves obtained by sweeping the threshold parameter for each method. The proposed Matrix detector achieves a substantially shorter detection delay than all three competitors at comparable false-alarm rates, even though the underlying intensity difference is not exactly low-rank.

*Table 3.* Simulation results for the dependent 10D PPP experiment on $[0,1]^{10}$ with the change occurring at time $1400$. We use $N_{\text{train}} = 1200$, $N_{\text{total}} = 1800$, and 100 Monte Carlo replications. ADD and SD are reported conditional on correct detection.

| Metric | Matrix | MMD | KIE | NN-CUSUM |
|---|---|---|---|---|
| False Alarm | 5% | 6% | 4% | 4% |
| Correct Detection | 95% | 88% | 90% | 89% |
| No Alarm | 0% | 6% | 6% | 7% |
| ADD (SD) | 24.21 (5.86) | 266.51 (76.57) | 284.33 (50.28) | 240.10 (55.81) |
| ADD (no-alarm removed) | 24.21 | 263.73 | 282.93 | 235.39 |

**Cumulative false-alarm rate under (M0).** We additionally evaluate the cumulative false-alarm rate over time on a 10-dimensional no-change scenario generated under the same temporally dependent pre-change model as above. Table 4 reports the cumulative false-alarm rates measured at the time grid $\{1200, 1400, \dots, 2200\}$, and Figure 5 shows the corresponding curves. The Matrix, MMD, and NN-CUSUM detectors all maintain a cumulative false-alarm rate within $5\%$ throughout, whereas the KIE detector's false-alarm rate drifts upward over time.

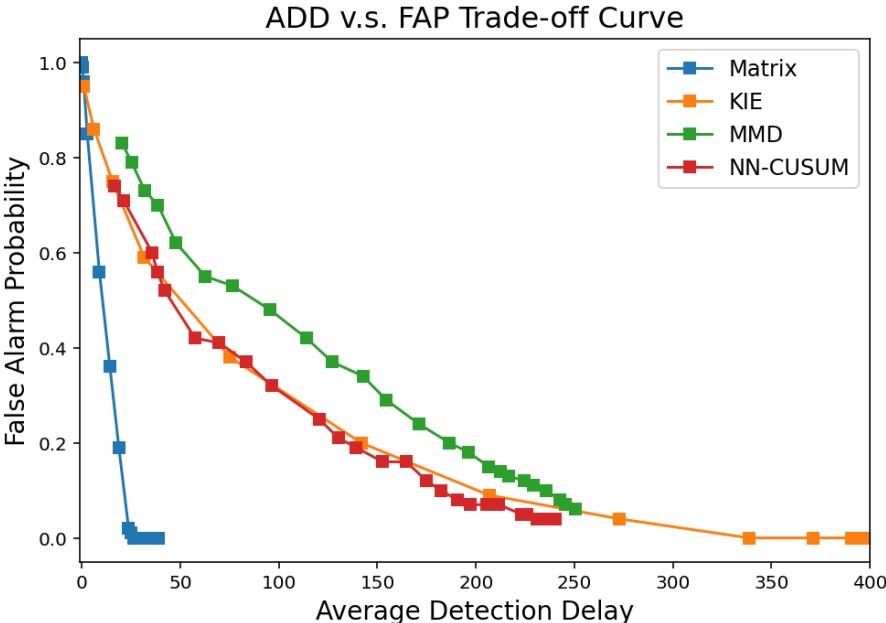

*Figure 4.* FAP vs. ADD trade-off comparison among the four detectors under the 10D non-low-rank setting.

*Table 4.* Cumulative false-alarm rates for the no-change dependent 10D PPP experiment, measured at time grids $\{1200, 1400, \ldots, 2200\}$. The data are generated under the same temporally dependent 10D PPP pre-change model as in Table 3.

| Time | MMD | KIE | Matrix | NN-CUSUM |
|------|-----|-----|--------|----------|
| 1200 | 0% | 0% | 0% | 0% |
| 1400 | 0% | 0% | 2% | 0% |
| 1600 | 1% | 3% | 2% | 1% |
| 1800 | 2% | 9% | 2% | 2% |
| 2000 | 2% | 11% | 3% | 2% |
| 2200 | 3% | 14% | 4% | 2% |

### E.2. Robustness to Rank Selection and Coordinate Partition

The Matrix detector requires two structural choices: the working rank $r$ used inside the restricted SVD in Algorithm 2, and the coordinate partition $[d] = \mathcal{I}_1 \cup \mathcal{I}_2$ used to form the matricized intensity. As clarified in Remark 2.5, Theorem 2.4 provides false-alarm and detection-delay guarantees for any partition that satisfies the approximate low-rank condition, and the empirical correlation-based criterion in Section 3.1 is a practical heuristic. Here we provide an empirical assessment of how sensitive the procedure is to these two choices.

We use the same dependent 10D PPP setting as in Section E.1. We evaluate four configurations of the Matrix detector:

- the default correlation-based partition with rank $r \in \{3, 5, 10\}$, and

- a randomly chosen coordinate partition with rank $r$ selected via the goodness-of-fit criterion of Section 3.1.

For each configuration, we sweep the threshold and record the resulting FAP–ADD trade-off. Figure 6 shows the four curves. The trade-off curves remain close to one another across all rank values and under the random-partition baseline, indicating that the proposed procedure is not sensitive to the specific rank choice or to the use of a non-correlation-based partition.

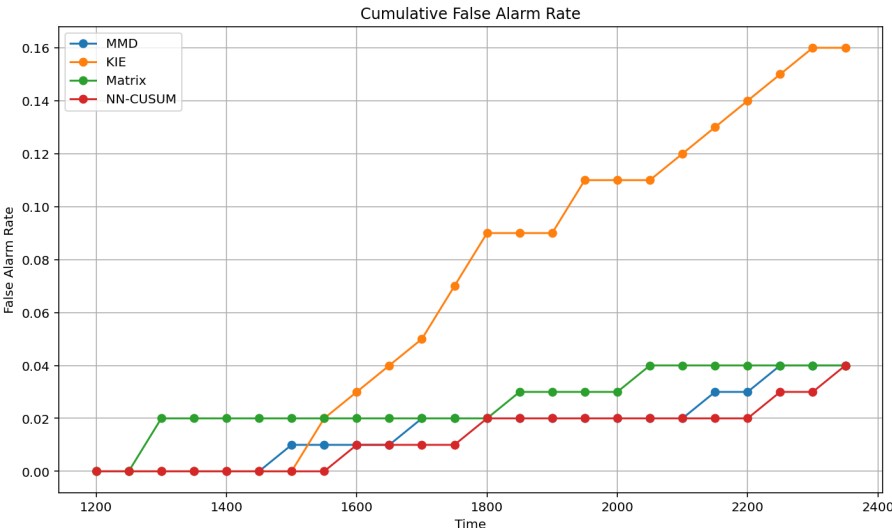

*Figure 5.* Cumulative false-alarm rate over time for the no-change dependent 10D PPP scenario. The Matrix, MMD, and NN-CUSUM detectors remain well within $5\%$ throughout the time horizon, while the KIE detector exhibits a noticeable upward drift.

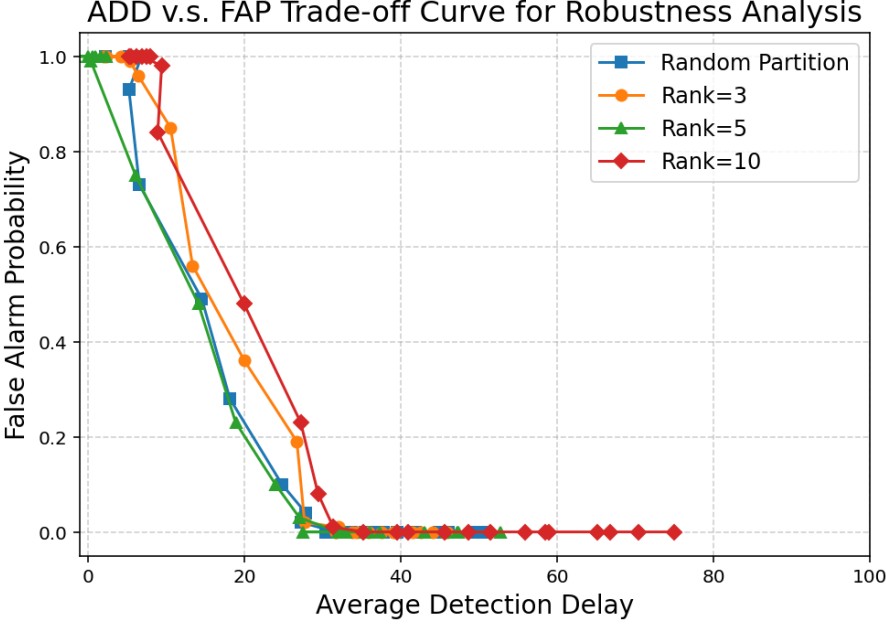

*Figure 6.* FAP vs. ADD trade-off curves for the Matrix detector under different rank choices ($r \in \{3, 5, 10\}$) and a random coordinate partition, in the dependent 10D PPP setting. The curves remain close to one another, indicating that the procedure is not sensitive to either choice.

