# OpenReview forum: "Online Change Point Detection for Multivariate Inhomogeneous Poisson Processes Time Series"
_ICML.cc/2026/Conference — ICML 2026 regular_

### Official Review · Reviewer_npRT · 2026-03-09

**Soundness:** 3
**Presentation:** 2
**Significance:** 3
**Originality:** 3
**Overall Recommendation:** 5
**Confidence:** 2

**Summary:**

This paper introduces a method to estimate a change-point in multidimensional Poisson processes that are dependent over time. The authors obtain finite-sample guarantees both on the probabilities of detecting a false and a true change-point. They demonstrate the efficiency of their procedure first on synthetic data and then on a real-world dataset from seismology.

**Compliance With Llm Reviewing Policy:**

Affirmed.

**Final Justification:**

The authors have provided additional empirical validation of their method during the rebuttal, both on synthetic and real-world datasets.
Therefore I increased my evaluation from 4 to 5.

**Key Questions For Authors:**

-  In the real-data illustration, the per-day aggregation of the data appears quite in contradiction with the assumption that the marginals of the process are Poisson processes, since it is likely that the locations of consecutive earthquakes will be dependent. I think it would be interesting to provide a numerical study with such aggregated data (and more generally of cases where the Poisson assumption is not verified) in order to see the effect on the estimation procedure

- I think that the main weakness of the paper is the limited numerical study, it would benefit from a thorough exploration of other scenarios (e.g. less abrupt changes in the intensities, misspecification of the model).

**Limitations:**

The paper mentions no limitation.

**Strengths And Weaknesses:**

Strengths
- the paper tackles a problem for which the literature appears quite scarce, in particular in an online and nonparametric framework
- the authors obtain strong theoretical guarantees for their estimator
- there are many potential applications of their method


Weaknesses
- the paper is technical and quite difficult to read, in particular because the authors refer to definitions that will appear later in the paper (for instance Equations 1 and 2 are cited in p1 while they are not defined until p3).
- the numerical study is restricted to two examples and for each only one parameters set, with no sensitivity analysis according to parameters variations.

---

> ### Author Rebuttal · Authors · 2026-03-28
>
> Thank you for your comments.
>
> Response to the Weaknesses.
>
> * The forward references in the introduction are only to standard notation, namely the Sobolev norm and the $\beta$-mixing coefficients, whose precise formulas are collected later in the notation section for completeness. These references are not needed to understand the high-level model statement, but we agree that they can make the paper harder to read on a first pass. In the revision, we will briefly define these quantities informally at first use, postpone equation-level cross-references until after the notation section, and add a short notation guide.
>
> * To the best of our knowledge, our work is the **first** to study online change detection for nonparametric multivariate PPP intensity functions under temporal dependence, and there is **no** existing literature dedicated to this problem. On the other hand, we adapt two popular methods for nonparametric **density** change point detection (KIE and MMD), for numerical comparison with our method. In the revision, we add the following additional numerical studies.
>   * A 10-dimensional simulation setting in which the intensity functions are not exactly low-rank. The results are summarized in Table R1, with the threshold calibrated by a block-permutation procedure. These results show that the proposed Matrix detector continues to outperform the competing methods in detection delay while maintaining a comparable false-alarm rate.
>   * We include a neural-network-based **density** change point detector, adapted from Gong et al. (2022), as an additional method.
>   * We add [robustness studies](https://anonymous.4open.science/r/Additional-numerical-studies-5800/plot/10D-robust.png) for rank selection and coordinate partitioning, which shows that the proposed method remains stable across different rank choices and under random coordinate splits.
>   * A second real-data analysis based on daily new COVID-19 case counts across all U.S. counties over 730 consecutive days from the beginning of 2020 to the end of 2021, using data from USAFacts. Each day is treated as one realization of a two-dimensional inhomogeneous Poisson process with longitude and latitude as coordinates. Our Matrix method raises an alarm on March 2, 2021, which is consistent with contemporaneous CDC reports documenting the rapid nationwide decline in new cases following the late-January 2021 peak.
>
> Responses to the Specific Questions.
>
> * This is a good point, and we conduct a new simulation study to address it. We first generate a temporally dependent 10D PPP time series on $[0,1]^{10}$, where the intensity changes at time $4200$ from the form $(1+\sin(\pi\sum_{j=1}^{10}x_j))$ to the form $(1+\cos(\pi\sum_{j=1}^{10}x_j))$, and temporal dependence is induced through the intensity scale. We set $N_{\mathrm{train}}=3600$ and $N_{\mathrm{total}}=5400$. Then, for every three consecutive time points, we combine the data into a single time instance to form the aggregated time series data. This mimics the setting in which Poisson data are aggregated and the model assumptions are no longer exactly satisfied. The simulation results are reported in the [ADD-FAP trade-off plot](https://anonymous.4open.science/r/Additional-numerical-studies-5800/plot/10D-aggregated.png), which shows that our proposed method continues to be robust among all the methods.
>
> * As mentioned in Point 2 of the Weaknesses response, we have conducted substantially more numerical experiments to further demonstrate the numerical performance of our method. These experiments examine settings with infinite-rank intensity functions, robustness with respect to tuning parameters, and a 10D PPP time series. We have also added an additional real-data example on COVID-19 case counts. We provide a [plot](https://anonymous.4open.science/r/Additional-numerical-studies-5800/plot/Covid.png) showing the difference in average daily new cases between the last two weeks of February and the first two weeks of March 2021, where blue indicates decreases and red indicates increases.
>
> **Table R1. Simulation results for the dependent 10D PPP experiment on $[0,1]^{10}$, where the intensity changes from the form ($1+\sin(\pi\sum_{j=1}^{10}x_j)$) to the  form ($1+\cos(\pi\sum_{j=1}^{10}x_j)$) at time $1400$ and temporal dependence is induced through the intensity scale. We use $N_{\mathrm{train}}=1200$, $N_{\mathrm{total}}=1800$, and 100 Monte Carlo replications.**
> | Metric | Matrix | MMD | KIE | NN-CUSUM |
> |---|---|---|---|---|
> | False Alarm | 5% | 6% | 4% | 4% |
> | Correct Detection | 95% | 88% | 90% | 89% |
> | No Alarm | 0% | 6% | 6% | 7% |
> | ADD (SD) | 24.21 (5.86) | 266.51 (76.57) | 284.33 (50.28) | 240.10 (55.81) |
> | ADD with no-alarm removed | 24.21 | 263.73 | 282.93 | 235.39 |

---

> > ### Author Rebuttal · Reviewer_npRT · 2026-04-02
> >
> > I thank the authors for the clarifications. I appreciate the additional empirical validation both on synthetic and real-world datasets.
> >
> > Overall, I am convinced by the answers of the authors during the rebuttal and I will increase my evaluation from 4 to 5.

---

> > > ### Author Response · Authors · 2026-04-03
> > >
> > > Thank you for your comments and acknowledgements!

---

### Official Review · Reviewer_vMSE · 2026-03-12

**Soundness:** 2
**Presentation:** 2
**Significance:** 2
**Originality:** 2
**Overall Recommendation:** 4
**Confidence:** 4

**Summary:**

This paper addresses the problem of online change-point detection in multivariate inhomogeneous Poisson point process (PPP) time series data exhibiting temporal dependence. Pointing out the limitations of existing methods that are either overly parametric or assume temporal independence, the authors propose a non-parametric matrix-based detector capable of operating under a geometric $\beta$-mixing environment. The primary goal is to minimize the detection delay when a post-training change occurs, while effectively controlling for false alarms.

**Compliance With Llm Reviewing Policy:**

Affirmed.

**Key Questions For Authors:**

1. Please address the concerns raised in the Weaknesses.
2. Why is a permutation-based threshold calibration valid in a model structure that explicitly allows for temporal dependence? Please clarify why it wouldn't be more appropriate to use a calibration procedure that does not compromise the dependence structure of the data (e.g., block bootstrap).
3. Please provide a sensitivity analysis showing how much the detection performance (especially ADD) degrades if the intensity function of the actual data does not satisfy the assumed degree of smoothness ($\gamma$), or if the underlying pattern of the data cannot be perfectly compressed into a low-rank matrix.
4. How does the restricted-SVD step respond if the post-change signal is not truly low-rank in reality, or if the rank is misspecified within the algorithm?
5. What is the clear logical rationale for choosing the method of minimizing empirical cross-group correlation during coordinate partitioning, and how sensitive is the model's performance to this specific partitioning choice?
6. What is the reason for excluding more recent online non-parametric detectors mentioned in the related works from the experimental baselines, aside from MMD and KIE?
7. A single earthquake dataset presents clear limitations in proving the general applicability of the proposed algorithm. Could you evaluate additional real-world datasets from heterogeneous domains mentioned in the introduction, such as crime occurrence patterns or epidemic surveillance, to demonstrate the model's scalability?
8. Please provide a clear guide on how to interpret the relative behavioral characteristics of the proposed method and the baseline models across various change regimes. A discussion is needed on the specific conditions under which the proposed model is overwhelmingly superior and the conditions under which it is not.

**Limitations:**

Please refer to the concerns raised in the Weaknesses and Key Questions.

**Strengths And Weaknesses:**

Strengths
1. While existing point process change point detection studies largely assume parameter-based models or temporal independence, this study successfully integrates realistic temporal dependence ($\beta$-mixing) into the model. In particular, deriving the Matrix Bernstein inequality (Theorem C.9) for PPP data following geometric $\beta$-mixing is a substantial academic contribution from the perspective of statistical learning theory.
2. By incorporating dynamic programming into the algorithm design, the computational complexity incurred when a new observation is added is maintained at a constant time level, independent of the length of the past data. This is a highly robust and practical advantage in real-world environments requiring real-time processing of high-dimensional streaming spatiotemporal data.
3. In the 3D and 4D synthetic data experimental settings, the proposed matrix detector demonstrated its practical effectiveness by significantly reducing the average detection delay (ADD) while maintaining a false alarm rate comparable to that of the MMD and KIE baselines.

Weaknesses
1. Although the paper explicitly assumes that the data follows geometric $\beta$-mixing, it employs a permutation method(randomly shuffling the training windows) for threshold calibration. This approach is methodologically inconsistent, as it risks destroying the temporal dependence structure that is central to the theoretical framework.
2. Coordinate partitioning is a core step of the algorithm; however, there is insufficient theoretical justification or ablation study showing why the heuristic of minimizing empirical cross-group correlation preserves or improves detection performance over alternative methods (e.g., random splitting).
3. Despite the excellent theoretical development, real-world data validation is restricted to a single earthquake dataset. Furthermore, by using only MMD and KIE as baselines, the paper omits performance comparisons with more recent online non-parametric detectors, making it difficult to judge its generalized superiority.

---

> ### Author Rebuttal · Authors · 2026-03-28
>
> Thank you for your comments.
>
> Response to the Weaknesses.
> 1. We replace permutation of individual training windows with a block-based permutation calibration, which preserves local dependence within each block and is more compatible with the $\beta$-mixing framework.
>
> 2. We add robustness studies for rank selection and coordinate partitioning.
>
> 3. To the best of our knowledge, our work is the **first** to study online change detection of nonparametric multivariate PPP intensity functions under temporal dependence. Therefore, there is **no** directly comparable existing baseline for our setting. On the other hand, we include a neural-network-based density change point detector, adapted from Gong et al. (2022), as an additional method.
> We add a second real-data analysis based on daily new COVID-19 case counts across all U.S. counties using data from USAFacts.
>
> Responses to the Specific Questions.
> 1. The concerns raised in the Weaknesses are addressed above.
>
> 2. In the revision, we use a block-based permutation procedure instead.
>
> 3. We add a 10-dimensional simulation setting, where the intensity functions are not exactly low-rank. The results are summarized in Table R1. We provide the corresponding FAR-ADD [trade-off plot](https://anonymous.4open.science/r/Additional-numerical-studies-5800/plot/10D.png). Regarding sensitivity to the smoothness parameter $\gamma$, it enters our procedure through the selection of the alarm-thresholding parameter rather than through a restrictive model-fitting step. As can be seen from the FAR-ADD trade-off plot, our method remains robust across a broad range of alarm-thresholding parameter values.
>
> 4. Our method does **not** require the intensity functions to be exactly low rank. Rather, the theory only requires that the tail singular values beyond rank $r$ be sufficiently small relative to the overall signal size, as formalized around Eq.(12). In addition, Lemma C.6 in our paper shows that this condition holds with a constant rank $r$ when the singular values decay polynomially (with degree $> 1/2$) or exponentially.  We also add a 10D simulation, where the intensity functions are **not** exactly low-rank. The results are summarized in Table R1.
>
> 5. The rationale for minimizing empirical cross-group correlation is to encourage a matrix representation with simpler cross-group interaction, which is more amenable to low-rank approximation. We also include a [robustness analysis](https://anonymous.4open.science/r/Additional-numerical-studies-5800/plot/10D-robust.png) for rank choice and coordinate partitioning, which shows that the method remains stable under different rank choices and under random coordinate splits.  The results indicate that the method is not sensitive to coordinate partition.
>
> 6. To the best of our knowledge, our work is the **first** to study online change detection of nonparametric multivariate PPP intensity functions under temporal dependence. Therefore, there is no directly comparable existing point-process baseline for our setting. In particular, existing  likelihood-based or scan-statistic  are not develop to leverage Poisson process structure with nonparametric intensity, and hence are not directly applicable here. On the other hand, we modify the method in Gong et al. (2022), which is a neural-network-based method for detecting nonparametric change points, and conduct a new experiment for 10D PPP time series. The results are summarized in Table R1 and the [threshold-tradeoff plot](https://anonymous.4open.science/r/Additional-numerical-studies-5800/plot/10D.png).
>
> 7. We add a second real-data analysis based on daily U.S. county-level COVID-19 case counts.
>
> 8. In terms of relative performance, our method is particularly advantageous for detecting change points in PPP time series in higher dimensions. The other methods, such as MMD, KIE, and NN-cusum detectors, are designed for density or distributional change-point detection and are therefore not well suited to the intensity change setting. Table R1 and the trade-off plot illustrate that our method has a clear advantage in the higher-dimensional setting.
>
> **Table R1. Simulation results for the dependent 10D PPP experiment on $[0,1]^{10}$, where the intensity changes from the form ($1+\sin(\pi\sum_{j=1}^{10}x_j)$) to the  form ($1+\cos(\pi\sum_{j=1}^{10}x_j)$) at time $1400$ and temporal dependence is induced through the intensity scale. We use $N_{\mathrm{train}}=1200$, $N_{\mathrm{total}}=1800$, and 100 Monte Carlo replications.**
> | Metric | Matrix | MMD | KIE | NN-CUSUM |
> |---|---|---|---|---|
> | False Alarm | 5% | 6% | 4% | 4% |
> | Correct Detection | 95% | 88% | 90% | 89% |
> | No Alarm | 0% | 6% | 6% | 7% |
> | ADD (SD) | 24.21 (5.86) | 266.51 (76.57) | 284.33 (50.28) | 240.10 (55.81) |
> | ADD with no-alarm removed | 24.21 | 263.73 | 282.93 | 235.39 |

---

> > ### Author Rebuttal · Reviewer_vMSE · 2026-04-03
> >
> > I appreciate the authors' efforts in preparing the rebuttal and providing additional robustness studies.
> > However, the response relies primarily on empirical observations rather than providing a rigorous theoretical justification for the Coordinate Partitioning heuristic, leaving a core methodological concern unresolved.
> > As this fundamental issue regarding the theoretical soundness of the algorithm has not been fully addressed, my score remains consistent.

---

> > > ### Author Response · Authors · 2026-04-03
> > >
> > > Thank you for this follow-up comment. We respectfully believe that the paper already provides a general and rigorous theoretical analysis for **any** coordinate partition satisfying the **approximate** low-rank condition in Eq. (12), and that the additional empirical studies are included only to evaluate the practical effectiveness of the partition-selection heuristic.
> > >
> > > First, the theory in the paper is already fairly general. Theorem 2.4 establishes false-alarm control and detection-delay guarantees for any coordinate partition, provided that under the chosen partition the matricized intensity satisfies the approximate low-rank condition in Eq. (12), namely that the singular values beyond rank $r$ are sufficiently small relative to the overall signal size. Thus, the theory does not rely on any special property of the partition-selection rule itself; it applies to all partitions satisfying this approximate low-rank structural condition.
> > >
> > > Second, this approximate low-rank condition is supported by standard functional approximation assumptions and is not restrictive. Lemma C.6 shows that it holds with a constant rank $r$ when the singular values decay polynomial (with exponent greater than $1/2$) or exponentially, both of which are standard assumptions in the literature; see, for example, Hall et al. (2006, The Annals of Statistics) and Raskutti et al. (2012, Journal of Machine Learning Research). It also covers common examples such as separable functions, additive functions, finite-rank expansions, and more general smooth or analytic interactions. In this sense, the condition is well motivated and extends beyond a narrow model class; related approximate low-rank structures also arise naturally in standard point process intensity models.
> > >
> > > Third, we clarify that the proposed empirical criterion, i.e. minimizing cross-group correlation, is not claimed to be an optimal partition-selection rule. Rather, it is a practical and numerically stable proxy for identifying a partition under which the cross-group interaction is simpler. This is analogous to the intuition behind functional PCA: when two coordinate groups are less strongly coupled, the corresponding cross-group operator is often well captured by a small number of dominant singular components. The additional simulations and real-data examples are included to support this practical point, not to replace theory. Moreover, the robustness studies also suggest that even random partitions can perform well as a simple default choice.
> > >
> > > In the revision, we will make this distinction more explicit: the theoretical guarantees are established conditional on a partition satisfying the approximate low-rank property, whereas the correlation-based rule is a practical heuristic for selecting such a partition. We will also strengthen the implementation discussion by considering data driven out-of-sample selection among candidate partitions, including exhaustive search when the dimension is moderate, to improve partition choice in practice.

---

### Official Review · Reviewer_tJS9 · 2026-03-13

**Soundness:** 3
**Presentation:** 3
**Significance:** 2
**Originality:** 3
**Overall Recommendation:** 5
**Confidence:** 4

**Summary:**

This paper studies online change-point detection for multivariate inhomogeneous Poisson point process (PPP) time series. The authors propose a method that represents the underlying PPP intensity function through an orthonormal basis expansion and maps it into a finite-dimensional matrix representation. Based on this representation, they construct a CUSUM-type statistic that compares estimated intensity matrices before and after candidate change points and apply a restricted SVD (low-rank projection) to enhance detection power. The resulting algorithm operates in a single-pass online manner with constant per-observation computational cost. The paper also provides theoretical guarantees on false alarm control and detection delay under β-mixing temporal dependence assumptions. Simulation experiments comparing the proposed Matrix detector with MMD-based and kernel intensity estimation (KIE) baselines show improved detection delay at comparable false alarm rates, and a real-data study on earthquake data demonstrates the practical applicability of the method.

**Compliance With Llm Reviewing Policy:**

Affirmed.

**Final Justification:**

The paper proposes a matrix-based sequential change detection procedure for multivariate Poisson point processes. The approach is technically interesting and introduces a novel perspective by exploiting low-rank structure in the intensity differences across dimensions. The method is supported by theoretical analysis and sequential detection guarantees, and the presentation of the algorithm and theoretical results is generally clear.

My main concerns in the initial review were related to the practical interpretation of the low-rank assumption, the limited empirical evaluation, and the lack of comparisons with existing point-process change detection approaches. The authors addressed these concerns in the rebuttal by providing additional explanations regarding the low-rank assumption and adding new experimental results. In particular, the additional simulation study that considers settings where the intensity difference is not exactly low-rank helps clarify the robustness of the proposed method. The authors also include a neural-network-based baseline and an additional real-data analysis using COVID-19 case data.

However, some concerns remain. The empirical evaluation is still somewhat limited, particularly in terms of the diversity of real-world point-process datasets. While the additional experiments in the rebuttal are helpful, the empirical evidence supporting the method’s broader applicability remains relatively modest.

Overall, the paper presents an interesting methodological contribution with solid theoretical support, and the rebuttal partially addressed my concerns. Taking into account the strengths of the proposed approach and the improvements provided in the rebuttal, I slightly increase my confidence in the work.

**Key Questions For Authors:**

Question 1: The proposed method relies on a low-rank SVD projection of the intensity difference. Could the authors provide more intuition or empirical evidence on when this low-rank structure is expected to hold in practice, particularly for real-world point process data?

Question 2: The empirical evaluation compares the proposed method with MMD and kernel intensity estimation baselines. Could the authors comment on how the method would compare with other point-process change detection approaches (e.g., likelihood-based or scan-statistic methods) that explicitly leverage Poisson process structure?

Question 3: The real-data experiment focuses on a single earthquake dataset. It would be helpful to know whether the method has been tested on additional datasets or scenarios, and how robust the results are across different types of point process data.

**Limitations:**

yes

**Strengths And Weaknesses:**

Strengths

The paper addresses the problem of online change-point detection for multivariate Poisson point process time series, which is an important yet relatively underexplored setting. The proposed method introduces a matrix representation of the intensity function combined with a restricted SVD statistic, leading to an efficient single-pass detection procedure. The theoretical analysis provides guarantees on false-alarm control and detection delay under temporal dependence assumptions. Empirical results in simulations and a real-data example demonstrate that the proposed method can achieve improved detection delay compared with the considered baselines.

Weaknesses:

The method may rely on structural assumptions such as an approximate low-rank representation of the intensity difference, which could limit its applicability in scenarios with more complex spatial patterns. The coordinate partition used to construct the matrix representation is selected heuristically, and the sensitivity of the detection performance to different partition choices is not thoroughly investigated. In addition, the empirical evaluation is relatively limited, with comparisons against only a small number of baselines and a single real-data example, which makes it harder to fully assess the robustness and practical advantages of the proposed approach.

---

> ### Author Rebuttal · Authors · 2026-03-28
>
> Thank you for your comments. We address the main concerns as follows.
>
> To the best of our knowledge, our work is the **first** to study online change detection for nonparametric multivariate PPP intensity functions under temporal dependence, and there is **no** existing literature dedicated to this problem. On the other hand, we adapt two popular methods for nonparametric **density** change point detection (KIE and MMD), for numerical comparison with our method. In the revision, we add the following additional numerical studies.
>   * A 10-dimensional simulation setting in which the intensity functions are not exactly low-rank. The results are summarized in Table R1, with the threshold calibrated by a block-permutation procedure. These results show that the proposed Matrix detector continues to outperform the competing methods in detection delay while maintaining a comparable false-alarm rate.
>   * We include a neural-network-based **density** change point detector, adapted from Gong et al. (2022), as an additional method.
>   * We add [robustness studies](https://anonymous.4open.science/r/Additional-numerical-studies-5800/plot/10D-robust.png) for rank selection and coordinate partitioning, which shows that the proposed method remains stable across different rank choices and under random coordinate splits.
>   * We add a second real-data analysis based on daily new COVID-19 case counts across all U.S. counties using data from USAFacts.
>
> Regarding the specific questions:
>
> Question 1: Our method does **not** require the intensity functions to be exactly low rank. Rather, the theory only requires that the tail singular values beyond rank $r$ be sufficiently small relative to the overall signal size, as formalized around Eq.(12). In addition, Lemma C.6 in our paper shows that this condition holds with a constant rank $r$ when the singular values decay polynomially (with degree $> 1/2$) or exponentially.  We also add a 10D simulation, where the intensity functions are **not** exactly low-rank. The results are summarized in Table R1 and the [threshold-tradeoff plot](https://anonymous.4open.science/r/Additional-numerical-studies-5800/plot/10D.png). We also add [robustness studies](https://anonymous.4open.science/r/Additional-numerical-studies-5800/plot/10D-robust.png) for rank selection and coordinate partitioning, which shows that the proposed method remains stable across different rank choices and under random coordinate splits.
>
> Question 2: To the best of our knowledge, our work is the **first** to study online change detection of nonparametric multivariate PPP intensity functions under temporal dependence. Therefore, there is no directly comparable existing point-process baseline for our setting. In particular, existing  likelihood-based or scan-statistic  are not develop to leverage Poisson process structure with nonparametric intensity, and hence are not directly applicable here. On the other hand, we modify the method in Gong et al. (2022), which is a neural-network-based method for detecting nonparametric change points, and conduct a new experiment for 10D PPP time series. The results are summarized in Table R1 and the [threshold-tradeoff plot](https://anonymous.4open.science/r/Additional-numerical-studies-5800/plot/10D.png).
>
> Question 3: We add a second real-data analysis based on daily new COVID-19 case counts across all U.S. counties over 730 consecutive days from the beginning of 2020 to the end of 2021, using data from USAFacts. Each day is treated as one realization of a two-dimensional inhomogeneous Poisson process with longitude and latitude as coordinates. Our Matrix method raises an alarm on March 2, 2021, which is consistent with contemporaneous CDC reports documenting the rapid nationwide decline in new cases following the late-January 2021 peak. For comparison, the competing methods detect the change later: MMD on March 4, 2021, NN-CUSUM on March 12, 2021, and KDE on April 17, 2021. To further illustrate this change, we provide a [plot](https://anonymous.4open.science/r/Additional-numerical-studies-5800/plot/Covid.png) showing the difference in average daily new cases between the last two weeks of February and the first two weeks of March 2021, where blue indicates decreases and red indicates increases.
>
> **Table R1. Simulation results for the dependent 10D PPP experiment on $[0,1]^{10}$, where the intensity changes from the form ($1+\sin(\pi\sum_{j=1}^{10}x_j)$) to the  form ($1+\cos(\pi\sum_{j=1}^{10}x_j)$) at time $1400$ and temporal dependence is induced through the intensity scale. We use $N_{\mathrm{train}}=1200$, $N_{\mathrm{total}}=1800$, and 100 Monte Carlo replications.**
> | Metric | Matrix | MMD | KIE | NN-CUSUM |
> |---|---|---|---|---|
> | False Alarm | 5% | 6% | 4% | 4% |
> | Correct Detection | 95% | 88% | 90% | 89% |
> | No Alarm | 0% | 6% | 6% | 7% |
> | ADD (SD) | 24.21 (5.86) | 266.51 (76.57) | 284.33 (50.28) | 240.10 (55.81) |
> | ADD with no-alarm removed | 24.21 | 263.73 | 282.93 | 235.39 |

---

> > ### Author Rebuttal · Reviewer_tJS9 · 2026-04-02
> >
> > The rebuttal addresses several of my concerns. In particular, the additional simulation study that considers non-exact low-rank settings and the robustness analysis help clarify the role of the low-rank assumption. The authors also include an additional neural-network-based baseline and a second real-data experiment. I slightly increase my score from 4 to 5.

---

> > > ### Author Response · Authors · 2026-04-03
> > >
> > > Thank you for your comments and acknowledgements!

---

### Official Review · Reviewer_ck27 · 2026-03-15

**Soundness:** 3
**Presentation:** 3
**Significance:** 2
**Originality:** 3
**Overall Recommendation:** 4
**Confidence:** 3

**Summary:**

The paper presents an online change-point detection method for multivariate inhomogeneous Poisson Point Process time series under the $\beta$-mixing temporal dependence assumption. The proposed approach maps the process to a finite-dimensional intensity matrix using an orthonormal basis expansion, and then applies a restricted SVD procedure to further reduce variance. The paper establishes theoretical guarantees and reports empirical results demonstrating that the method is both efficient and accurate.

**Compliance With Llm Reviewing Policy:**

Affirmed.

**Final Justification:**

My concerns have been addressed.

**Key Questions For Authors:**

1. Could the authors further explain why the last equality holds in Eq. 10?

2. Is the fact that the computational cost does not grow with the length of the past time series really an advantage? I think this is natural for window-based methods.

3. In Algorithm 1, $M$ is initialized but does not appear later in the procedure. What role does $M$ play here?

4. Were the baselines used in the experiments designed for PPP? Why were the methods mentioned in the related work on PPP not used as baselines?

5. Since the delay is set to the length of the time series after the change if No Alarm, it seems natural that the baselines would have larger ADD because their no-alarm rate is higher than that of the proposed method. What are their ADD values in the cases where an alarm is actually reported? This statistic could also be valuable.

6. Are there any experimental results for scenario $M_1$ with no change point?

7. Could the authors explain Figure 3 in more detail? Could it be visualized as a time series instead of as an average statistical summary?

**Limitations:**

Yes.

**Strengths And Weaknesses:**

*Soundness*

The paper allows for temporal dependence in Poisson point process time series through a $\beta$-mixing dependence assumption, which goes beyond the existing work about Poisson point processes. The proposed algorithm is supported by theoretical guarantees. Although I did not verify the proofs, the theoretical development appears sound overall. However, the parametric dependence assumption is still relatively strong.

The simulation studies and a real-world dataset demonstrate the effectiveness of the method in comparison with two baseline approaches. The authors evaluate performance using multiple metrics, and the proposed algorithm outperforms both baselines. However, the range of baselines is somewhat limited, and the simulation settings could be more comprehensive. In particular, the current experiments only consider process dimension $3$ and a fixed change point at time point 1200.

*Presentation*

The paper is well organized and the overall framework is clearly presented. The plots are also clear.

*Significance*

The paper addresses an important and practical problem: change-point detection in inhomogeneous Poisson point process time series with temporal dependence.

*Originality*

The proposed algorithm is novel in handling temporal dependence. The paper establishes a new Matrix Bernstein inequality for $\beta$-mixing Poisson point process time series. The paper also establishes finite-sample guarantees on the false alarm probability and shows that true change points can be detected within a short interval with high probability.

---

> ### Author Rebuttal · Authors · 2026-03-28
>
> Thank you for your comments. We address the main concerns as follows.
>
> * Our dependence assumption is **not** parametric: the intensities are modeled via nonparametric Sobolev classes, and temporal dependence is characterized through $\beta$-mixing, which does not specify a finite-dimensional time series model.
>
> * To the best of our knowledge, our work is the **first** to study online change detection for nonparametric multivariate PPP intensity functions under temporal dependence, and there is **no** existing literature dedicated to this problem. On the other hand, we adapt two popular methods for nonparametric **density** change point detection (KIE and MMD), for numerical comparison with our method. In the revision, we add the following additional numerical studies.
>   * A 10-dimensional simulation setting in which the intensity functions are not exactly low-rank. The results are summarized in Table R1, with the threshold calibrated by a block-permutation procedure. These results show that the proposed Matrix detector continues to outperform the competing methods in detection delay while maintaining a comparable false-alarm rate.
>   * We include a neural-network-based **density** change point detector, adapted from Gong et al. (2022), as an additional method.
>   * We add [robustness studies](https://anonymous.4open.science/r/Additional-numerical-studies-5800/plot/10D-robust.png) for rank selection and coordinate partitioning, which shows that the proposed method remains stable across different rank choices and under random coordinate splits.
>   * A second real-data analysis based on daily new COVID-19 case counts across all U.S. counties over 730 consecutive days from the beginning of 2020 to the end of 2021, using data from USAFacts. Each day is treated as one realization of a two-dimensional inhomogeneous Poisson process with longitude and latitude as coordinates. Our Matrix method raises an alarm on March 2, 2021, which is consistent with contemporaneous CDC reports documenting the rapid nationwide decline in new cases following the late-January 2021 peak.
>
> Regarding the specific questions:
> 1. The last equality in Eq. 10 follows from the fact that vector integration   and therefore  matricization (operator $\mathcal{M}$) of the intensity function is linear. Consequently,
> $\\mathcal{M}(\\lambda^{\\ast}) - \\mathcal{M}(\\lambda_{a}^{\\ast}) = \\mathcal{M}(\\lambda^{\\ast} - \\lambda_{a}^{\\ast})$.
> 2. We will change our wording on computational cost.
> 3. The quantity $M$ is the maximal basis resolution, which determines the size $M^p \times M^q$ of each estimated intensity matrix $\widehat{\mathcal M}^{(i)}$.
> 4. The methods cited in our related-work section address different settings, including parametric Hawkes/network models (Wang et al. 2023; Zhang et al. 2023), model-based offline spatio-temporal change point estimation (Zhao et al. 2019), and offline multiple change point detection for 1D point processes (Dion et al. 2023), and are therefore not directly comparable to our online nonparametric multivariate PPP setting.
> 5. We would like to clarify that the ADD values reported in Tables 1 and 2 of the submitted paper are already computed conditional on correct detection, that is, only over runs in which an alarm is raised after the change point. For comparison, Table R1 in this response reports ADD both no-alarm included and no-alarm removed, which are nearly identical.
> 6. We add a 10D no-change simulation. The cumulative false alarm rates are summarized in Table R2, which shows that Matrix, MMD, and NN-CUSUM remain well controlled over time (all within 5%).
> 7. We add a [time series of heatmaps](https://anonymous.4open.science/r/Additional-numerical-studies-5800/plot/earth-time-seris.png) for the estimated yearly average earthquake intensity.
>
> **Table R1. Simulation results for the dependent 10D PPP experiment on $[0,1]^{10}$, where the intensity changes from the form ($1+\sin(\pi\sum_{j=1}^{10}x_j)$) to the  form ($1+\cos(\pi\sum_{j=1}^{10}x_j)$) at time $1400$ and temporal dependence is induced through the intensity scale. We use $N_{\mathrm{train}}=1200$, $N_{\mathrm{total}}=1800$, and 100 Monte Carlo replications.**
> | Metric | Matrix | MMD | KIE | NN-CUSUM |
> |---|---|---|---|---|
> | False Alarm | 5% | 6% | 4% | 4% |
> | Correct Detection | 95% | 88% | 90% | 89% |
> | No Alarm | 0% | 6% | 6% | 7% |
> | ADD (SD) | 24.21 (5.86) | 266.51 (76.57) | 284.33 (50.28) | 240.10 (55.81) |
> | ADD with no-alarm removed | 24.21 | 263.73 | 282.93 | 235.39 |
>
> **Table R2. Cumulative false alarm rates for the no-change dependent 10D PPP experiment at time grids 1200, 1400, $\cdots$, 2200. The data are generated under the same temporally dependent 10D PPP pre-change model as in Table R1.**
>
> | Time | MMD | KIE | Matrix | NN-CUSUM |
> |---|---|---|---|---|
> | 1200 | 0% | 0% | 0% | 0% |
> | 1400 | 0% | 0% | 2% | 0% |
> | 1600 | 1% | 3% | 2% | 1% |
> | 1800 | 2% | 9% | 2% | 2% |
> | 2000 | 2% | 11% | 3% | 2% |
> | 2200 | 3% | 14% | 4% | 2% |

---

> > ### Author Rebuttal · Reviewer_ck27 · 2026-04-04
> >
> > Thank you to the authors for the response and the additional experimental results.
> >
> > Most of my questions have been addressed. I still have two points for further clarification:
> >
> > 1. My concern regarding the fixed true change point, specifically “a fixed change point at time point 1200,” has not yet been addressed. Is the method sensitive to the location of the true change point? My concern is whether the method can still handle a change point effectively when it occurs before enough samples have been observed.
> >
> > 2. I still do not fully understand what Figure 3 is intended to illustrate. The estimated change point is reported as June 2009, so I am confused about why Figure 3 shows two subfigures with June 2010 used as the separation point. In addition, why do the corresponding heatmaps in the provided link differ from those shown in the main paper? Please correct me if I have misunderstood.

---

> > > ### Author Response · Authors · 2026-04-06
> > >
> > > Thank you for your follow-up questions.
> > >
> > > 1. To investigate this issue, we conducted an additional simulation study in the dependent $10$-dimensional PPP setting. We varied the training size as $N_{\mathrm{train}} = \{150,200,250\}$ and the true changepoint as $\mathrm{cp} = \{N_{\mathrm{train}}+30, N_{\mathrm{train}}+50, N_{\mathrm{train}}+70\}$,
> > > while fixing the total sample size at $N_{\mathrm{total}}=500$. Thus, in all settings the change occurs shortly after the training period ends, directly testing whether the method remains effective when only a limited number of monitored pre-change observations are available after training. By varying $N_{\mathrm{train}}$, we also assess robustness to different amounts of pre-change calibration data.
> > >
> > > The results are summarized below.
> > >
> > > **$N_{\mathrm{train}}=150$ and $N_{\mathrm{total}}=500$. Each entry reports false alarm rate / correct detection rate / ADD on correct detection (SD).**
> > >
> > > | $\mathrm{cp} - N_{\mathrm{train}}$ | Matrix | MMD | KIE | NN-CUSUM |
> > > |---|---|---|---|---|
> > > | 30 | 5% / 95% / 25.71 (3.68) | 15% / 13% / 66.00 (76.35) | 4% / 47% / 44.00 (44.20) | 7% / 70% / 93.57 (87.35) |
> > > | 50 | 7% / 93% / 27.42 (4.23) | 13% / 13% / 67.38 (63.70) | 2% / 48% / 44.83 (41.50) | 18% / 38% / 93.95 (82.18) |
> > > | 70 | 4% / 96% / 29.39 (3.09) | 19% / 8% / 85.75 (94.96) | 6% / 51% / 37.69 (34.68) | 19% / 40% / 94.25 (71.50) |
> > >
> > > **$N_{\mathrm{train}}=200$ and $N_{\mathrm{total}}=500$. Each entry reports false alarm rate / correct detection rate / ADD on correct detection (SD).**
> > >
> > > | $\mathrm{cp} - N_{\mathrm{train}}$ | Matrix | MMD | KIE | NN-CUSUM |
> > > |---|---|---|---|---|
> > > | 30 | 4% / 96% / 24.11 (3.35) | 12% / 19% / 61.58 (61.32) | 3% / 77% / 39.38 (47.36) | 5% / 55% / 84.27 (65.67) |
> > > | 50 | 4% / 96% / 25.46 (2.94) | 11% / 9% / 84.44 (66.45) | 5% / 44% / 39.82 (43.29) | 10% / 40% / 98.50 (64.91) |
> > > | 70 | 5% / 95% / 27.79 (3.06) | 10% / 8% / 81.50 (67.73) | 4% / 46% / 53.87 (46.54) | 14% / 41% / 100.37 (67.72) |
> > >
> > > **$N_{\mathrm{train}}=250$ and $N_{\mathrm{total}}=500$. Each entry reports false alarm rate / correct detection rate / ADD on correct detection (SD).**
> > >
> > > | $\mathrm{cp} - N_{\mathrm{train}}$ | Matrix | MMD | KIE | NN-CUSUM |
> > > |---|---|---|---|---|
> > > | 30 | 4% / 96% / 22.82 (3.33) | 5% / 19% / 49.68 (53.32) | 9% / 70% / 27.06 (19.02) | 6% / 84% / 60.65 (54.43) |
> > > | 50 | 4% / 96% / 24.59 (2.89) | 12% / 8% / 48.25 (51.77) | 2% / 49% / 36.53 (29.19) | 5% / 35% / 93.86 (47.78) |
> > > | 70 | 2% / 98% / 25.89 (3.03) | 8% / 9% / 45.56 (51.49) | 4% / 47% / 46.72 (37.88) | 5% / 59% / 77.88 (42.69) |
> > >
> > > Across all nine settings, the proposed Matrix detector is highly stable. Its false-alarm rate ranges from 2% to 7%, its correct-detection rate ranges from 93% to 98%, its no-alarm rate is 0% throughout, and its ADD remains between 22.82 and 29.39.
> > > By comparison, all competing methods are worse in both correct-detection rate and ADD on correct detections.
> > > Overall, these results suggest that, the proposed Matrix method is not sensitive to the exact location of the changepoint and continues to perform well even when the change occurs soon after monitoring begins. Moreover, for the proposed Matrix detector, a larger training size generally leads to better performance, with the most visible improvement appearing in the ADD, while the false-alarm and correct-detection rates remain consistently strong.
> > >
> > > At the same time, we would like to clarify the scope of our method. Our current theory is designed for the setting in which the change occurs \emph{after} the training period used for calibration. If the true change occurs before $N_{\mathrm{train}}$, then the training sample itself is contaminated by post-change observations, which falls outside the setting covered by our current guarantees. More broadly, assuming access to a sufficiently large pre-change sample for estimating the pre-change distribution is common and practically meaningful in online changepoint detection; see, for example, Wang and Xie (2024, Wiley Interdisciplinary Reviews).
> > >
> > >
> > >
> > > 2. For your second question, we acknowledge that there are typos in the figure titles. The estimated changepoint is indeed June 2009. The intended interpretation of Figure 3 is that the upper subfigure shows the estimated average intensity from 06/2007 to 06/2009 (before the estimated changepoint), while the lower subfigure shows the estimated average intensity from 07/2009 to 07/2011 (after the estimated changepoint). The heatmaps in the provided link differ because they represent estimated yearly average earthquake intensities, computed over July to June of the following year, rather than the two before/after averaged intensity surfaces shown in the paper. We will correct the figure titles in the revision.

---

### Decision · Program_Chairs · 2026-04-30

**Decision:**

Accept (regular)

**Comment:**

This paper is among the first to develop an approach for online change detection in multivariate Poisson processes with nonparametric intensity functions that exhibit temporal dependence. The proposed approach parameterizes the multivariate intensity function via low-rank matrices and induces temporal dependence via β-mixing. The reviewers all appreciated the significance and novelty of the paper. While the reviewers were initially split on a number of key points, such as the appropriateness of the low-rank assumption, or the adequacy of the empirical evaluations, the authors assuaged many of these concerns during the discussion. One reviewer remained concerned about the sensitivity of the proposed approach to the choice of coordinate partition despite supplemental experiments provided during the discussion period that suggested robustness to such choices. However, the authors convincingly argued that the paper establishes general theory for under any partition, and that the heuristics used to select one in practice are not central to the paper's main contributions. The authors should revise the paper to make this point more clear, but I think this paper is in good shape otherwise.